# Three dimensional classification of dislocations from single projections

Tore Niermann [1,2] ✉, Laura Niermann [1,2] & Michael Lehmann[1]

Many material properties are governed by dislocations and their interactions. The reconstruction of the three-dimensional structure of a dislocation network so far is mainly achieved by tomographic tilt series with high angular ranges, which is experimentally challenging and additionally puts constraints on possible specimen geometries. Here, we show a way to reveal the three dimensional location of dislocations and simultaneously classify their type from single 4D scanning transmission electron microscopy measurements. The dislocation's strain field causes inter-band scattering between the electron's Bloch waves within the crystal. This scattering in turn results in characteristic interference patterns with sufficient information to identify the dislocations type and depth in beam direction by comparison with multi-beam calculations. We expect the presented measurement principle will lead to fully automated methods for reconstruction of the three dimensional strain fields from such measurements with a wide range of applications in material and physical sciences and engineering.

Dislocations and their interaction are responsible for a wide range of material properties, ranging from strengthening of metals and alloys[1] to efficiency in semiconductor laser devices[2]. Thus, knowledge of the three dimensional topology of dislocation networks is crucial for material and interface engineering[3]. A two-dimensional projection of dislocation networks can be readily obtained by conventional (scanning-) transmission electron microscopy (S/TEM) images[4].

The three dimensional topology of dislocation networks is currently mainly investigated by means of tomographic methods. The first reconstruction was done using X-ray topography tomography, however at low resolution[5]. Tilt-tomography of weak beam dark field images within the TEM allows the investigation of dislocation networks at 5 nm resolution[6]. By a combination of tilt-tomography STEM with Fourier filtering even higher resolutions might be achievable[7], even though the resolution limits of such an approach are debated[8]. These tomographic techniques have in common that a large number of images must be obtained with high tilt ranges (typically 25 to 200 micrographs with angular ranges > 120 degrees), furthermore they are often troubled by the missing wedge problem and dynamical scattering effects[9].

In thin GaN-samples of below 20 nm thickness, the classification and depth determination of dislocations was successfully demonstrated using multislice-ptychography[10] and depth sectioning[11]. However, in thicker specimen dynamical diffraction effects will cause an increasing challenge for these methods.

For classification of dislocation types the projected line vector of a dislocation is readily obtained from conventional S/TEM images, and the direction of the Burgers vector $\mathbf{b}$ can be determined by the famous $\mathbf{g} \cdot \mathbf{b} \neq 0$ criterion for reflections $\mathbf{g}$[1,12,13]. From convergent beam electron diffraction (CBED) patterns of defocused probes (convergent beam imaging) even the Burgers vector's length and sign can be determined from higher order Laue zone line splittings at the dislocation line[14].

Instead of mixing diffraction and imaging information like done in convergent beam imaging we instead use 4D-STEM[15] to collect zeroth order Laue zone (ZOLZ) CBED pattern with non-overlapping disks for each scan point with focused probes. These measurements are performed under illumination conditions, where the resulting CBED patterns are governed by dynamical diffraction. Under such conditions the propagation of electrons through the crystal is affected by inhomogeneities $\frac{\partial}{\partial z}(\mathbf{g} \cdot \mathbf{u})$ of the displacement field $\mathbf{u}$ in beam direction $z$ and their depth ($z$-position) within the specimen[16–18]. These inhomogeneities are for instance local shears and rotations of the lattice caused by the dislocation.

[1]Technische Universität Berlin, Institut für Optik und Atomare Physik, Straße des 17. Juni 135, 10623 Berlin, Germany. [2]These authors contributed equally: Tore Niermann, Laura Niermann. ✉e-mail: tore.niermann@tu-berlin.de

The effects of these inhomogeneities on the electron beam propagation can be understood in the Bloch wave picture[16]. In an unstrained crystal, the Bloch waves propagate undisturbed through the crystal with different longitudinal wave vector components depending on the Bloch wave band. Eventually, the interference between the individual Bloch waves at the exit surface of the specimen is observed in the diffraction pattern. The different longitudinal Bloch wave vector components cause the well known beating of the intensity with crystal thickness $t$ (Pendellösung). Inhomogeneities of the displacement field cause the electrons to scatter longitudinally between Bloch wave bands (inter-band scattering). Due to the interference of the Bloch waves at the exit surface this redistribution of Bloch wave excitations is detectable in the diffraction pattern. Also a lateral scattering of the Bloch waves on the respective dispersion surfaces of a Bloch wave band (intra-band scattering) occurs. However, this intra-band scattering has only a minor effect on the resulting diffraction patterns compared to the inter-band scattering. Since the interference of the Bloch waves at the exit surface not only depends on the difference of the longitudinal components of the Bloch wave vectors but also on the distance traveled within the crystal, the resulting interference is also sensitive on the depth of the inhomogeneity within the specimen. A CBED pattern allows the inspection of these Bloch wave interferences for several incident beam directions at once.

Nevertheless, a fundamental ambiguity of electron scattering for centro-symmetric scattering geometries exists[19]: displacements fields with inhomogeneities, which exhibit a mirror symmetry with respect to the specimen midplane in beam direction will result in the same diffraction pattern.

For a given Burgers vector and line vector of a dislocation the displacement field can be analytically calculated in simple cases like for isotropic elasticity[20], or numerically in general cases[21]. For a known strain field and a given electron probe position relative to the dislocation, the expected CBED patterns can be efficiently simulated by means of multi-beam calculations[12].

In this work, we show a method to uniquely determine the type and three dimensional position of dislocations (with the exception of the mentioned midplane symmetry). For this, spatial variations of CBED patterns with distance from the dislocation are extracted from a 4D-STEM measurement and these patterns are compared to calculated patterns. We demonstrate this method by determining the depth and type of dislocations within a hetero-epitaxial films of wurtzite-type GaN on a sapphire substrate.

## Results

### Specimen overview

The specimen is a wurtzite-type GaN film grown in [0001]-direction on a sapphire substrate. The lattice mismatch between the GaN layer and the substrate result in dislocations threading through the film in the growth direction[22]. Perfect dislocations within this material system are those of the hexagonal lattice and are characterized by Burgers vectors $\mathbf{b}$ of $\mathbf{a} = \frac{1}{3}\langle\bar{1}\bar{1}20\rangle$, $\mathbf{c} = \langle0001\rangle$, or $\mathbf{a}+\mathbf{c} = \frac{1}{3}\langle\bar{1}\bar{1}23\rangle$, with $\mathbf{a}$ and $\mathbf{c}$ corresponding to the base vectors of the lattice[20,23].

A region roughly 750 nm above the interface between the GaN-buffer and the sapphire substrate was investigated under two systematic row conditions, namely the (0002) and ($2\bar{1}\bar{1}0$) systematic rows only in order to demonstrate the method for different excitation conditions (see Supplementary Fig. 5 for a larger area image overview). Annular dark field (ADF) images of the investigated region for both systematic-row conditions are shown in Fig. 1b, d. This region was selected since it exhibits several dislocations of different types. These dislocations are emerging threading dislocations rooted in the interfacial misfit. Within this region a dislocation (marked A in the figure) with a line vector along the [0001] direction is observable, which is only strongly visible in the (0002) systematic row. The dislocations B, C, and D with line vectors roughly 45 degrees inclined to the [0001]-direction

are only strongly visible in the ($2\bar{1}10$) systematic row. Additionally, two basal stacking faults can be seen (E). Using the $\mathbf{g}\cdot\mathbf{b}$ criterion the set of possible Burgers vectors for these dislocations can already be reduced to: ±[0001] for dislocation A, and $\pm\frac{1}{3}[2\bar{1}\bar{1}0]$, $\pm\frac{1}{3}[1\bar{2}10]$, $\pm\frac{1}{3}[11\bar{2}0]$ for dislocations B, C, and D. Partial dislocations can be ruled out since the dislocations are not connected to other extended defects. Since the Burgers and line vectors for dislocation A are parallel this dislocation is of screw type, while dislocations B, C, and D are of mixed type. In the following we will further investigate dislocation A and B. The investigation of dislocations C and D is similar to the analysis of dislocation B. Please note, that the circular features present in the right half of the images originate from carbon contamination during the microscopy session and are not caused by crystalline defects in the image.

### Study of dislocation A

As sketched in Fig. 1a and further elaborated in the Methods section, a two dimensional $(q,x)$-plane was obtained from the 4D-STEM data. Please note, that the systematic row and thus the reciprocal space direction $q$ can be chosen independently from the spatial $x$ direction. For dislocation A the $(q,x)$-plane is obtained in the spatial dimension $x$ along the red arrow in Fig. 1b and in the diffraction dimension $q$ along the (0002)-systematical row (red arrow in Fig. 1c). For this plane the spatial dimension $x$ is roughly oriented along the [$2\bar{1}\bar{1}0$]-direction, i.e. perpendicular to the line-vector of the dislocation, with its origin $x = 0$ nm at the intersection with the dislocation line. Its reciprocal space dimension $q$ corresponds to the diffraction vector along the (0002)-systematic row. A similar $(q,x)$-plane was obtained for dislocation B.

The resulting $(q,x)$-plane for dislocation A is shown in Fig. 2a. Along the $q$-direction the CBED patterns of the 5 innermost reflections of the systematic row are clearly visible as separated intervals. Along the $x$-direction the variation of these CBED patterns in dependence of the distance $x$ to the dislocation can be seen. For positions sufficient far away from the dislocation the pattern resembles the CBED pattern of an unstrained crystal (see Supplementary Fig. 9). Such a behavior is seen in Fig. 2a, where the patterns further away ($|x| \gtrsim 35$ nm) from the dislocation become constant with $x$. The difference between the patterns for $x \lesssim -35$ nm and $x \gtrsim 35$ nm can be explained by a bending of the specimen caused by the far field of the dislocation's strain field. Closer to the dislocations core ($|x| \lesssim 20$ nm) more complicated features are observed, which are caused by the stronger strain in the near field of the dislocation. At the dislocation core itself a discontinuity of the patterns is observable.

Figure 2b shows the calculated $(q,x)$-plane for the parameters best matching this experimental dataset. Details on the calculation can be found in the Methods section. A very good agreement between the experimental and simulated $(q,x)$-planes can be found. Typical CBED features like the periodic fringes within each reflection occur at similar points and with similar intensities. Smaller deviations are mainly found in the upper region with $x < -25$ nm and are probably caused by the strain field of dislocation D. Also minor deviations are found at the projected dislocation core. However, these are expected due to inaccuracies of the simulation at the core (see Methods section). The best match was found for a specimen thickness of $t = 132$ nm, a depth of the dislocation core of $d = 55$ nm, an incident beam tilt of $\tau = 3.7$ mrad, a Burgers vector of $\mathbf{b} = [000\bar{1}]$, and line vector of [0001]. A Burgers or line vector of opposite sign would result in a $(q,x)$-plane with a flipped $x$-direction.

More insight in the quality of the match can be gained from the mean squared error (MSE), i.e. the mean squared intensity difference between experiment and calculation, which is shown in Fig. 2c for different specimen thicknesses $t$ and depths of the dislocation $d$. The minimum for the matching parameters is quite distinct: the MSE of the second lowest minimum was 42% larger than the MSE of the global minimum. The general dissimilarities of the simulated $(q,x)$-planes for different parameters of depth and thickness, can be seen from Supplementary Movies 1 and 2.

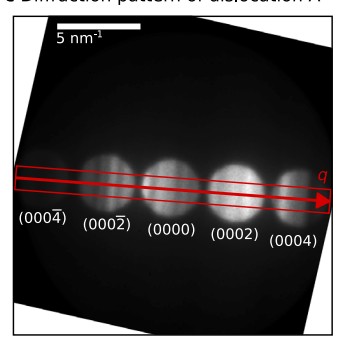

**Fig. 1 | Dataset overview. a** Scheme of the acquisition and evaluation process of the 4D-dataset. The specimen is tilted into a systematic row condition. The electron beam scans in within the $(x, y)$-plane over the specimen and for each scan position a diffraction pattern is acquired. The direction of the systematic row defines the reciprocal space direction $q$. The diffraction patterns with the same $x$-distance to the dislocation are averaged along the perpendicular $y$-direction. The intensities in these averaged diffraction patterns are further averaged perpendicular to the systematic row-direction (in $q'$-direction). In this way, for every $x$-position a $q$-profile is obtained, which results in intensities $I(q, x)$ within a $(q, x)$-plane. **b** Annular dark field (ADF) image in (0002) systematic row conditions with evaluated area marked by red rectangle (the red arrow marks the spatial direction $x$ for dislocation A). **c** Diffraction pattern averaged over all scan coordinates within the red rectangle of **a** (the red arrow marks the reciprocal space direction $q$ for dislocation A). **d** ADF image in (2$\overline{1}\overline{1}$0) systematic row conditions of the same region with evaluated area marked by red rectangle (the red arrow marks the spatial direction $x$ for dislocation B). **e** Diffraction pattern averaged over all scan coordinates within the red rectangle of **d** (the red arrow marks the reciprocal space direction $q$ for dislocation B). The crystal-directions are indicated in images **b** and **d**. The reflections are indicated in the diffraction patterns **c** and **e**. The diffraction pattern in **c** and **e** have been flipped and rotated to match the scan coordinate system of **b** and **d**.

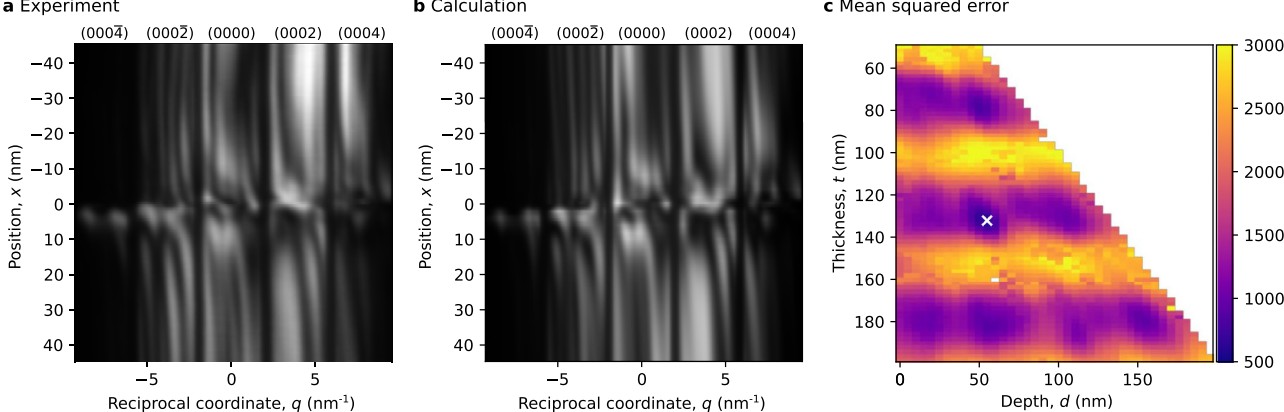

**Fig. 2 | Comparison of $(q, x)$-planes for dislocation A under (0002)-systematic row condition. a** Experimental intensities. **b** Simulated intensities for a Burgers vector of **b** = [000$\overline{1}$]. **c** Mean squared error map between experimental and calculated intensities for different values of specimen thickness $t$ and dislocation depth $d$ (uncolored points indicate untested points or diverged fits, white cross labels parameters used in **b**). Profiles through the mean squared error map can be found in Supplementary Fig. 8.

## Study of dislocation B

Figure 3a shows the $(q, x)$-plane of dislocation B obtained from the dataset in (2$\overline{1}\overline{1}$0) systematic row. The $x$-direction of this plane is indicated by the arrow in Fig. 1d and its $q$-direction is oriented along the systematic row in diffraction space (see Fig. 1e). The best matching calculation was found for a specimen thickness of $t = 172$ nm, a depth of the dislocation core of $d = 85$ nm, an incident beam tilt of $\tau = 0.83$ mrad, a Burgers vector of **b** = $\frac{1}{3}[\overline{1}2\overline{1}0]$, and line vector parallel to

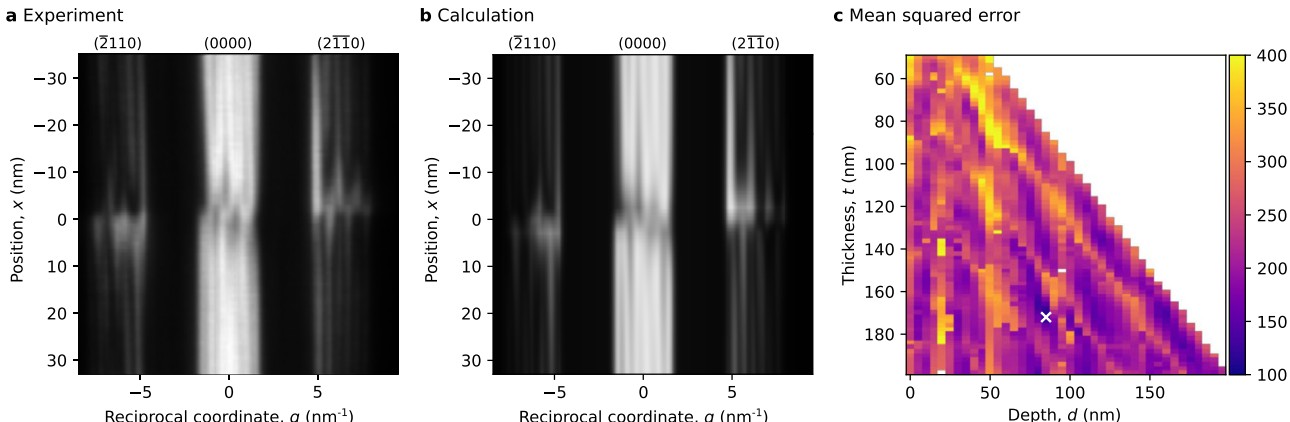

**Fig. 3 | Comparison of $(q, x)$-planes for dislocation B under $(2\bar{1}\bar{1}0)$-systematic row condition. a** Experimental intensities. **b** Simulated intensities for a Burgers vector of $\mathbf{b} = \frac{1}{3}[\bar{1}2\bar{1}0]$. **c** Mean squared error map between experimental and calculated intensities for different values of specimen thickness $t$ and dislocation depth $d$ (uncolored points indicate untested points or diverged fits, white cross labels parameters used in **b**). Profiles through the mean squared error map can be found in Supplementary Fig. 8.

$[14\,\bar{7}\,\bar{7}\,15]$. The simulated $(q, x)$-plane for these parameters is shown in Fig. 3b. While experiment and calculation in generally match well, some differences especially close to the core ($|x| \lesssim 5$ nm) in the (0000)-beam can be found, which we attribute to the inaccurate simulation of the effects of the strong strain field close to the core. The map of the MSE between experimental and calculated $(q, x)$-planes in Fig. 3c shows that the minimum is not as distinct as in the case for dislocation A: the MSE of the second lowest minimum was 18% larger than the MSE of the global minimum (see also the parameter sweep in the Supplementary Movies 3 and 4).

A comparison of the experimental $(q, x)$-plane with calculations for a Burgers vector of $\mathbf{b} = \frac{1}{3}[2\bar{1}\bar{1}0]$ shows significant differences (see Supplementary Fig. 6). However, the comparison with the calculations for an Burgers vector of $\mathbf{b} = \frac{1}{3}[11\bar{2}0]$ shows a similar matching calculation for a dislocation depth of $d = 90$ nm (see Supplementary Fig. 7). This similarity corresponds to the aforementioned mid-plane ambiguity of electron diffraction, since the $(2\bar{1}\bar{1}0)$-systematic row is along a centro-symmetric direction, the directions $[\bar{1}2\bar{1}0]$ and $[11\bar{2}0]$ only have an opposing component in beam direction, thus result in a displacement field with flipped components in beam direction, and the depth of both dislocation core are approximately located at similar distances but in opposing directions from the midplane. Since the centro-symmetry is broken in the $[0001]$ direction, no such mid-plane ambiguity exists for dislocation A.

## Discussion

From the MSE maps in Figs. 2c and 3c it can be seen that several local minima exists. However, the global minimum was always sufficient well identified. The reported thicknesses were verified by electron holography[24] as an alternative method for thickness determination (see Supplementary Note 2). Within Supplementary Note 1 we additionally demonstrated the described technique on a mechanically deformed Aluminum sample as alternative material system, where dislocation depth and type could also be successfully identified. The described method even successfully identifies the depth although with a less distinct minimum, when the spatial extents of the $(q, x)$-planes in the present example is reduced to as low as 11 nm, compared to the 89 nm in Fig. 2 (see Supplementary Note 3).

In the second example above the global minima was less prominent than in the first example. We attribute this to the weaker scattering within $(2\bar{1}\bar{1}0)$-systematic row compared to (0002) and the thicker specimen, the former leads to less distinct CBED patterns, while the latter leads to finer CBED patterns. For thicker specimen (like also observed for the Aluminum example in the Supplementary Note 1)

also inelastically scattered electrons, which experienced a plasmon loss, contribute significantly to the experimental CBED patterns[25]. These inelastically scattered electrons will result in more blurred patterns within the diffraction disk[26] as well as a diffuse background, which, however, is mitigated by the subtraction of the empirically modeled background.

We expect that the method can be improved in future: a possible improvement might be the use of zero-loss filtered diffraction patterns, which exhibit a higher quality for quantitative comparison of the elastic signal[27]. Also the a-priori knowledge that dislocations lines either end at the surfaces or in interactions with other defects can be used in combination with tracing the dislocation line at several positions.

The ambiguity due to mid-plane symmetry can not be resolved by the presented method for centro-symmetric cases. Replacing the sample with a sample mirrored at the mid-plane, however, should make no difference for nearly all practical applications, as due to the centro-symmetry the material properties remain the same. Even in cases where the symmetry is broken by the geometry of the sample, e.g. by interfaces, this direction would not be placed in beam direction in a typical S/TEM experiment. A 3D model of dislocation networks might still be obtained by selecting consistent depths and types for neighboring regions of a dislocation, e.g. by a suitable regularization.

In the presented cases we could determine the depth of the dislocation within the step-size of 5 nm used in the calculations, the specimen thickness could be determined with a similar precision. Beside the quality of the comparison metric this precision is also determined by the difference in longitudinal components of the Bloch wave vectors as well as number of Bloch waves excited, both depend on the material and the excitation conditions, such that even higher precisions might be possible. The accuracy of the depth also depends on the strain fields and structure factors used as input into the calculations. The isolated atom approximation and the absorptive optical potentials limit the accuracy of the multi-beam calculations. The strain fields used in the calculations assumed infinite volumes and isotropic elasticity. Nevertheless, we consider these approximations to be accurate enough for the claimed 5 nm precision. However, for dislocations close to the specimen surfaces the calculations are not accurate enough, and more complex strain models must be investigated that included for example relaxation effects.

The ability to three dimensionally locate the dislocation within a specimen, while simultaneously classify their type provides an extremely powerful way for the investigation of dislocation networks. From the 3D structure of dislocation networks more information about

crystal plasticity and the spatial interactions of dislocation with interfaces may be obtained. This 3D classification is possible with a single 4D-STEM measurements within the limitations of the **g·b** criterion. This technique is in principle not limited to the systematic row, and might be also performed close to zone axis conditions, where more Burgers vector orientations can be covered in a single measurement.

Compared to tomographic methods a single 4D-STEM measurement might be performed much faster and with less dose, such that dislocations networks can be investigated in more beam sensitive materials and also on more dynamic conditions, e.g. during a in-situ heating/cooling experiment. For instance, future in-situ experiments may study the evolution of a dislocation network during annealing or under external load.

Furthermore, we expect this measurement technique can be developed further into an automated determination and classification scheme using a pre-calculated data base of dislocation's fingerprints and incorporation of further a-priori knowledge about the material system. Such an automation might also be a potential application for machine learning approaches, where neural networks have been trained on fingerprint databases.

## Methods

### Experiment

The specimen slab was prepared with surfaces close to the $(01\bar{1}0)$ crystal planes by a conventional cross-section TEM preparation method consisting out of mechanical grinding followed by ion milling until electron transparency. For 4D-STEM measurements the specimen was rotated by roughly 4 degrees from the $[0\bar{1}10]$ zone axis into the respective systematic conditions.

For the 4D-STEM measurements the region was scanned on a $256 \times 256$ point grid with sampling steps of 0.78 nm. For each scan point the central part of the diffraction pattern was recorded with a Quantum Detector MerlinEM single chip detector on a $256 \times 256$ point grid with 0.75 nm$^{-1}$ sampling. The datasets were obtained using a JEOL GrandARM F2 microscope operated at 300 kV in Cs-corrected STEM-mode with an illumination semi-convergence angle of $\Theta = 3.3$ mrad and a dwell time of 1.3 ms.

### Data processing

For further analysis the information in the region of interest of the 4D-STEM dataset was reduced to a two dimensional $(q, x)$-plane, which has a spatial $x$-dimension and reciprocal space $q$-dimension. This reduction of the dataset is sketched in Fig. 1a. For the dataset evaluated for dislocation A, the corresponding directions are shown Fig. 1b, c: in the spatial dimensions all points of the 4D dataset with scan coordinates within the sub-region marked by the red rectangle in Fig. 1b are averaged in the direction perpendicular to the line indicated by the red arrow. In the diffraction dimension all points with reciprocal space coordinates within the sub-region marked by the red rectangle in Fig. 1c are averaged in the direction perpendicular to the line indicated by the red arrow. For the dataset in $(2\bar{1}\bar{1}0)$ systematic row condition the corresponding direction of the investigated $(q, x)$-plane are shown in Fig. 1d, e. This data reduction to a $(q, x)$-plane corresponds to the common operation of obtaining one dimensional profiles from two dimensional images. However, here this operation is performed twice, once in the spatial dimensions and once in the diffraction dimensions of the 4D dataset.

For display purposes the diffraction patterns in Fig. 1c, e were mirrored and rotated to match the orientation of the scanning grids in Fig. 1b, d. The ADF images in Fig. 1 were calculated from the 4D datasets by integrating over the scattering angles in the range from 3.3 mrad to 10.2 mrad for each scan point.

### Calculations

In order to attribute the Bloch wave interference patterns visible in the $(q, x)$-planes to specific dislocation types and dislocation depths multi-

beam scattering simulations are performed, where all beams of the respective systematic rows within $\pm 30$ nm$^{-1}$ were considered. These simulations are based on the numerical propagation of the Darwin–Howie–Whelan (DHW) equations along the beam direction ($z$-direction, here assumed parallel to the $[01\bar{1}0]$ crystal direction) and are performed within the column approximation[12,16]. The propagation is performed using a 4th-order Runge–Kutta scheme[28] with a step size of 0.1 nm.

The Fourier coefficients of the specimen's potential (including absorption effects as optical potential) are calculated for an unstrained GaN-crystal within the isolated atom approximation from parameterized data[29]. The effect of the displacement field $\mathbf{u}(x, z)$ is modeled as additional position dependent geometric phase of these coefficients[30]. All simulations were performed with the line vector along the $y$-direction, such that the displacement field is constant in that direction. Within the column approximation scattering due to lateral changes of the displacement is ignored. Thus the resulting intensities only carry a parametric dependence to the lateral position $x$. However, the effects of displacement field inhomogeneities in the $z$-direction are fully included.

Even though the DHW-equations describe the dynamical diffraction in a plane-wave base their numerical propagation also can be used to correctly model the inter-band scattering of Bloch-waves. In the Bloch wave picture the column approximation corresponds to the restriction to inter-band scattering (opposed to intra-band scattering)[16]. However, following the discussion above we consider the effect of lateral scattering on the resulting diffraction patterns to be negligible except for the uttermost core of the dislocation.

For the simulations the well-known analytical displacement fields of dislocations in elastically isotropic media are used[20]. We consider the effects of the elastic anisotropy to be negligible within the validity limits of the calculation. The direction of the line vectors of the dislocations and the possible types of Burgers vector are taken from the corresponding dark field images. Simulations in dependence of the dislocation core depth $d$ (measured from entrance surface) were performed for all Burgers vectors compatible with the observed **g·b** case.

Using $k$ to characterize the component of the incident wave vector along the systematic row, the simulation returns the diffraction intensities $I_g(x, -k; t, d)$ for all beams $g$ included for a given specimen thickness $t$ and dislocation depth $d$. The intensities $I(x, q; t, d)$ corresponding to the intensities obtained in a $(q, x)$-plane from the scanning convergent beam experiments are eventually given by

$$I(x, q;\, t, d) = N \sum_g I_g(x, q + k_0 - g;\, t, d) \ \text{ for } g \text{ with } \lambda |q + k_0 - g| < \Theta,$$

(1)

where $N$ is the total intensity in the beam, $k_0$ is the lateral component of the central beam's wave vector along the systematic row and is used to characterize the incident beam's tilt $\tau = \lambda k_0$. Furthermore, $\Theta$ is the illumination's semi-convergence angle and $\lambda$ the vacuum wave length. The additional restriction regarding $g$ mimics the effect of the illumination aperture. To match the grid of the experimental data the simulation data was bi-linearly interpolated within the $(q, x)$-plane. Please note, that in all presented experiments the semi-convergence angle is smaller than the Bragg angle, such that the CBED disks of the individual diffraction do not overlap and no interference effects between the beams need to be considered.

### Comparison

Beside the Bragg reflections also a diffuse background can be found within the experimental data between the reflections. This diffuse background originates from scattering at the amorphized surfaces due to specimen preparation, from carbon contamination within the microscope and from inelastic scattering. For a quantitative

comparison of the experimental data with the calculations this diffuse background is empirically modeled as a broad Gaussian intensity distribution, which is added to the calculated intensities. The Gaussian is adjusted in height and width such, that it remains below the intensity minima found between the reflections in the experimental data. Eventually, the calculated intensity data is convoluted with the point spread function of the detector[31], which was calculated from its modulation transfer function as measured separately under the same detection settings with the knife-edge method.

The experimental and calculated data were quantitatively compared using the mean squared error (MSE) as metric. The MSE is the average of the squared intensity differences. Please note, that the MSEs for different experimental datasets are in general not comparable with each other due to the different overall electron dose. The MSE was minimized under variation of specimen thickness $t$, depth of dislocation $d$, total intensity $N$, beam tilt $\tau$ and the exact positions of the dislocation and diffraction pattern center in the experimental data. Specimen thickness $t$ and dislocation depth $d$ were tested for all relevant values with 2 nm steps in thickness and 5 nm steps for depth. The other parameters were numerically minimized for a given set of ($t$, $d$) using the Broyden–Fletcher–Goldfarb–Shanno method implemented numerical Python package scipy[32].

All calculations and data processing were performed using Python and the PyCTEM toolkit. Further information about calculation times can be found in Supplementary Note 4.

## Data availability
The experimental data generated in this study have been deposited in the Zenodo repository with the identifier (https://doi.org/10.5281/zenodo.10458023)[33].

## Code availability
A GitHub repository containing the code used in the analysis is available (https://github.com/niermann/match_qx)[34].

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

## Acknowledgements

This work was funded by the Deutsche Forschungsgemeinschaft (DFG, German Research Foundation) within projects 492463633 (L.N.) and 403371556 (M.L.). We thank Sören Selve for providing us the Aluminum sample.

## Author contributions
L.N. and T.N. conceived the idea and designed the experiment. L.N. conducted the experiment and analyzed the data. T.N. conducted the simulations and developed the algorithms used for simulation and data

analysis. L.N. and T.N. wrote the paper. M.L. contributed suggestions and revised the manuscript. All authors read and commented on the manuscript.

## Funding

## Competing interests
The authors declare no competing interests.
