## [Peer Review File · Nature Communications]

Three dimensional classification of dislocations from single projectionsREVIEWER COMMENTS

Reviewer #1 (Remarks to the Author):

In this paper the authors report on detecting depth and type of dislocations from 4D-STEM series taken with non-overlapping CBED discs. The authors apply data reduction by appropriate averaging in real and reciprocal space and compare the intensity maps as a function of scan position x and reciprocal space coordinate q with simulation. The authors refine the tilt of the incident electron beam with respect to the crystal's zone axis, the specimen thickness, depth of the dislocation line and Burgers vector. The paper is extremely interesting as it proposes and demonstrates a method that avoids experimentally demanding tomographic tilt series. The suggested method is new and sound and the paper is written clearly and convincingly. The following minor comments could be considered:

1. The distance between the investigated regions around dislocations A and B marked in Figures 1a and b is approximately 100 nm, the measured difference in specimen thickness is 40 nm, which appears quite large. Did the authors check the local specimen thickness with a different method? If this thickness gradient is real, is the assumption of a constant thickness along the scanned regions (red rectangles) adequate?
2. The authors used the numerical propagation of the DHW equations. Would a multislice-calculation also include intra-band scattering?
3. Can the authors comment on the expected influence of using anisotropic elasticity theory to calculate the displacement fields?
4. Typos:
Line 309 "minima" should be "minimum"
Lines 891 and 900 "intensities intensities"

Reviewer #2 (Remarks to the Author):

In this paper the authors present a quantitative method for determining the type, depth, and dislocation line in 3 dimensions from a single 4D-STEM scan along a single projection. The quantitative work is carefully done, and the result is a novel take on a method for 3 dimensional strain determination from 4D-STEM.

The precision of the dislocation depth measurement is stated to be 5nm based on the localization of a global minimum of a summed squared differences metric. However, many local minima exist, as noted by the authors. Can the authors provide a more robust statistical error bar on the depth measurement based on the quantitative squared differences metric for the confidence in the global minimum vs other of the many local minima?

I have some significant doubts about the general applicability of this method for a wide variety of systems. The demonstration case shown involves a highly perfect crystal with only a few dislocations visible in a field of view of a few hundred nanometers. While the approximately 5nm depth resolution appears to have been fairly convincingly demonstrated for this system, the authors claim a fairly wide applicability to metals and other systems. However, most other systems will exhibit much higher dislocation and other defect densities (sometimes by orders of magnitude) than epitaxially grown semiconductors.

For example, the q - x plots are evaluated over a range of 80 nm. If the nearest neighbor dislocation were less than 40 nm away on either side, how much would this affect the precision of the result? It also appears that diffraction patterns were integrated over a ~ 40 nm window in the y -direction, which will have similar implications about the maximum dislocation densities that can be studied. What is the depth precision if the width of the q - x window is reduced to, for example, 20nm instead of 80 nm? What about a 10nm window? What if there is more than one dislocation (or another defect) present

along the beam axis?

What is the minimum sample thickness for this method to work reliably? For example, would it be applicable to nanoparticle quantum dots (or other nanoparticle systems) in the <20nm range?

Overall, I believe that this work is of high quality, and suitable for publication with a more quantitative representation of the depth precision. However, I believe that concerns about the general applicability of this method to systems with even slightly higher defect densities (which will encompass most real materials systems), suggest that eventual publication should take place in a more specialized journal than Nature Communications.

Reviewer #3 (Remarks to the Author):

Niermann et al. classify and determine the 3D position of dislocations on the nanoscale using four-dimensional scanning transmission electron microscopy (4D-STEM) and electron microscopy simulations. Tomographic data acquisition is much more tedious than single projection imaging, both experimentally and in post-processing (each projection needs to be aligned). Although dislocations have been determined in three dimensions using ADF STEM [1] and electron ptychography [2] at the nanoscale without tilting, the noteworthy improvement of the work of Niermann et al. is the applicability of this method to thick STEM samples (>100nm). By acquiring 4D-STEM data over different dislocations, plotting diffraction-real-space maps (q-x maps) and comparing to multi-beam calculations via summed squared difference maps, the z-position of two different dislocations are determined in a GaN sample on a sapphire substrate.

I believe this work is well constructed and self-contained, and will be a welcome addition to the electron microscopy literature. Sample thickness limitations are one of the greatest challenges in transmission electron microscopy, and this technique accounts for this quite well. However, I am doubtful of the universality of this technique. Extensive calculations are required to find the local minimum for the sample thickness and depth of dislocation. These simulations may take considerable time, and would need to be performed for every material to be studied. If these simulations are easily implemented over reasonable time scales and, additionally, applicable to many crystal types, this would be worthy of publication in Nature Communications. For now, it seems like extensive work needs to be done to get to this point. In conclusion, I would not recommend publication in Nature Communications, but believe that a revised draft of this work would certainly be worthy of publication in Scientific Reports or a Microscopy Journal.

I have provided some specific feedback and comments below.

[1] Yang, H., Lozano, J. G., Pennycook, T. J., Jones, L., Hirsch, P. B., & Nellist, P. D. (2015). Imaging screw dislocations at atomic resolution by aberration-corrected electron optical sectioning. *Nature Communications*, 6(1), 7266.

[2] Gilgenbach, C., Chen, X., Xu, M., & LeBeau, J. (2023). Three-dimensional Analysis of Nanoscale Dislocation Loops with Multislice Electron Ptychography. *Microscopy and Microanalysis*, Volume 29, Issue Supplement_1, Pages 286–287.

Comments

1. I believe Figure 1 should include a pipeline of the analytical process. For example, use text to emphasize which directions are being averaged. Then in Fig 1d insert a subfigure with the q-x map determined from Fig 1a. In its current form, it is difficult to interpret the scientific workflow from Figures 1-3 and the supporting text.

2. Can you comment on the 3D precision/accuracy of this technique? The authors mention that they can identify the depth location within the limits of their simulation step sizes. Was this step size

chosen because of an existing limit, or could you have used a smaller step size and located the dislocation more accurately?

3. The advantage of this technique over ADF optical sectioning and ptychography is the relaxed sample thickness limitations. What is the maximum thickness that enables faithful depth determination? Also, what is the minimum thickness?

4. Some of the figure labels are incorrect (figure 2 instead of figure 1)

5. I believe the time taken to complete the simulations will be a critical factor in determining whether this approach will be implemented throughout the microscopy community. How long did the simulations take to run?

6. I found the supplementary movies to be very helpful in understanding the findings. I suggest explicitly referring to these movies (i.e. Movie S1, S2, etc) in a revised draft.

7. "From the squared difference maps in Fig. 2c and 3c it can be seen that several local minima exists. However, the global minima always was sufficient well identified. This was confirmed by visual inspection of the other possible candidates." – minima exist*, global minimum*, sufficiently*, was always*

Could this 'visual inspection' step be replaced with a quantitative comparison metric?

Response letter

First, we thank the reviewers for taking the time and effort to provide constructive comments, which have been very helpful in making improvements to the manuscript.

Please find below our detailed response to the individual reviewers' comments. Beside the changes described in response to the individual points below, we also revised the manuscript in the following points:

1) Two of the reviewers raised concerns for the applicability of the method to a wider range of materials. In response to these concerns the editor suggested to demonstrate the method on other materials. Within the supplementary information we added the application of the method to a mechanically deformed Aluminum specimen containing a dislocation network (Supplementary Data 2).

For the Aluminum sample we observed an ambiguity in the data, which we hadn't previously noticed. Further literature study (see references in the revised manuscript) revealed that this ambiguity is a physically fundamental property of electron scattering: in centrosymmetric materials (or cases where the systematic row itself is in a centrosymmetric crystal direction) the electron scattering within the specimen results in identical diffracted beam intensities, if the specimen's displacement field is antisymmetric with respect to the specimen's mid-plane in electron beam direction. Accordingly, the method can not discriminate between dislocation geometries, which result in such antisymmetric displacement fields. For instance, in the demonstrated Aluminum sample, cases of a flipped Burger's vectors component along the beam direction with a simultaneously mirrored distance of the dislocation with respect to the specimen's mid-plane could not be discriminated.

Eventually this ambiguity just means, that the sign of the z-direction can not be recovered for centrosymmetric cases. This is only one bit of information, which can not be resolved by the presented method.

This ambiguity of electron scattering is correctly reproduced by the method described in our manuscript, which furthermore demonstrates the reliability of the method. In the GaN the ambiguity was previously not recognized since dislocation A was investigated in the non-centrosymmetric (0002) systematic row (where the ambiguity doesn't exist) and dislocation B was located very close to the mid-plane.

We don't see this ambiguity as a limitation for the presented method, as we explain in an additional paragraph in the discussion of the revised manuscript:

The ambiguity due to mid-plane symmetry can not be resolved by the presented method for centro-symmetric cases. Replacing the sample with a sample mirrored at the mid-plane, however, should make no difference for nearly all practical applications, as due to the centro-symmetry the material properties remain the same. Even in cases where the symmetry is broken by the geometry of the sample, e. g. by interfaces, this direction would not be placed in beam direction in a typical S/TEM experiment. A 3D model of dislocation networks might still be obtained by selecting consistent depths and types for neighboring regions of a dislocation, e. g. by a suitable regularization.

We added a paragraph to the introduction:

Nevertheless, this technique can not resolve a fundamental ambiguity of electron scattering for centro-symmetric scattering geometries [Koprucki2022]: displacements fields, which exhibit an antisymmetric mirror symmetry with respect to the specimen mid-plane (i. e. $u_z(z) = -u_z(t - z)$ with specimen thickness t) will result in the same diffraction pattern.

And describe the observation of this ambiguity in the last paragraph of the results:

A comparison of the experimental (q,x)-plane with calculations for a Burgers vector of $b=1/3 [2 -1 -1 0]$ shows significant differences (see Supplementary Figure 2). However, the comparison with the calculations for an Burgers vector of $b=1/3 [1 1 -2 0]$ shows a similar matching calculation for a dislocation depth of $d=90\text{nm}$ (see Supplementary Figure 3). This similarity corresponds to the aforementioned mid-plane ambiguity of electron diffraction, since the $(2 -1 -1 0)$ -systematic row is along a centro-symmetric direction, the directions $[-1 2 -1 0]$ and $[1 1 -2 0]$ only have an opposing component in beam direction, thus result in a displacement field with flipped components in beam direction, and eventually the dislocation core is located in a opposing distance from the mid-plane. Since the centro-symmetry is broken in the $[0001]$ direction, no such mid-plane ambiguity exists for dislocation A.

2) We changed the figure of merit for the comparison between the experimentally obtained (q,x)-planes and the simulated ones to the mean square error (MSE), which is the mean instead of the sum of the squared differences. Both metrics are proportional to each other, however the MSEs are still comparable if the investigated region is changed in size (see answer to remark 2.2). The displayed squared distance maps in the figures, the figure captions, and references to them in the main text were adjusted accordingly. The second to last paragraph of the methods sections, which introduces this metric, was also changed accordingly:

The experimental and calculated data were quantitatively compared using the mean squared error (MSE) as metric. The MSE is the average of the squared intensity differences. Please note, that the MSEs for different experimental datasets are in general not comparable with each other due to the different overall electron dose. The MSE was minimized...

3) Within the revised discussion the additionally introduced supplementary data is referenced:

From the MSE maps in Fig. 2c and 3c it can be seen that several local minima exists. However, the global minimum was always sufficient well identified. The reported thicknesses were verified by electron holography [Lehmann2016] as an alternative method for thickness determination (see Supplementary Data 2). Within Supplementary Data 1 we additionally demonstrated the described technique on a mechanically deformed Aluminum sample as alternative material system, where dislocation depth and type could also be successfully identified. The described method even successfully identifies the depth although with a less distinct minimum, when the spatial extents of the (q,x)-planes is reduced to as low as 11 nm, compared to the 89 nm in Fig. 2 (see Supplementary Data 3).

3) The diffraction pattern for dislocation B (previous extended data item 2) is now part of the main text as Fig. 1e.

4) In the first paragraph of the results section, the set of possible a-type Burgers vectors were stated with the wrong length ($1/3$ was missing), which was now corrected. Within the analysis always the

correct Burgers vectors were used. The length was correctly given in all other occasions of the original manuscript (introduction, captions, ...)

5) The Burgers vector in the caption of supplementary figure 3 (previously extended data item 4) was erroneously stated as $1/3 [1 -2 10]$. The comparisons were done for a Burgers vector of $1/3 [1 1 -2 0]$, as now is correctly stated, and the figure always showed the results for the (now corrected) vector. A Burgers $1/3 [1 -2 10]$ results in a flipped x-axis with respect to the $1/3 [1 1 -2 0]$ case.

6) Typos (including those mentioned by the reviewers) in the main text have been fixed.

7) Previous Extended Data items are now added as Supplementary Figures to the Supplementary Information document, references to these items were adjusted accordingly.

8) In order to comply with Nature Communications style guide, we distilled an abstract from the original first paragraph:

Many material properties are governed by dislocations and their interactions. The reconstruction of the three-dimensional structure of the network so far is mainly achieved by tomographic tilt series with high angular ranges, which is experimentally challenging and additionally puts constraints on possible specimen geometries. Here, we show a new way to reveal the three dimensional location of dislocations and simultaneously classify their type from single 4D scanning transmission electron microscopy measurements. The dislocation's strain field causes inter-band scattering between the electron's Bloch waves within the crystal. This scattering in turn causes characteristic interference patterns with sufficient information to identify the dislocations type and depth in beam direction by comparison with multi-beam calculations. We expect the presented measurement principle will lead to fully automated methods for reconstruction of the three dimensional strain fields from such measurements with a wide range of applications in material and physical sciences and engineering.

The first paragraph of the introduction was adjusted accordingly, to avoid repetition:

Dislocations and their interaction are responsible for a wide range of material properties, ranging from strengthening of metals and alloys [Hull2011] to efficiency in semiconductor laser devices [Nakamura1998]. Thus, knowledge of the three dimensional topology of dislocation networks is crucial for material and interface engineering [Sutton1996]. A two-dimensional projection of dislocation networks can be readily obtained by conventional (scanning-) transmission electron microscopy (S/TEM) images [Hirsch2006].

Reviewer #1

“In this paper the authors report on detecting depth and type of dislocations from 4D-STEM series taken with non-overlapping CBED discs. The authors apply data reduction by appropriate averaging in real and reciprocal space and compare the intensity maps as a function of scan position x and reciprocal space coordinate q with simulation. The authors refine the tilt of the incident electron beam with respect to the crystal's zone axis, the specimen thickness, depth of the dislocation line and Burgers vector. The paper is extremely interesting as it proposes and demonstrates a method that avoids experimentally demanding tomographic tilt series. The

suggested method is new and sound and the paper is written clearly and convincingly. The following minor comments could be considered:”

Remark 1.1: *“The distance between the investigated regions around dislocations A and B marked in Figures 1a and b is approximately 100 nm, the measured difference in specimen thickness is 40 nm, which appears quite large. Did the authors check the local specimen thickness with a different method? If this thickness gradient is real, is the assumption of a constant thickness along the scanned regions (red rectangles) adequate?”*

We checked the thickness by using the phase information obtained by electron holography (EH) in this region. The thicknesses resulting from the EH measurements (A: 139.5 +/- 9.7 nm, B: 166 +/- 11 nm) are within the errors of the thicknesses reported in the manuscript (A: 132 +/- 5 nm, B: 172 +/- 5 nm).

We added a section to the supplementary information (Supplementary Data 2) describing this alternative thickness determination.

These thickness gradients slightly compromises the precision of the described method, which can be seen from the improved mean squared error for smaller averaged regions (see our answer to remark 2.2). However, in the current examples the gradient seems to sufficiently average out. In future applications of the method a thickness gradient along the x-direction of the (q,x)-planes could be additionally considered within the simulated (q,x)-plane. In principle the extent of the averaging region in the spatial direction perpendicular to x could also be chosen much smaller. Eventually this extent depends on the shot-noise within the specimen and the amount of noise acceptable for the comparison.

Remark 1.2: *“The authors used the numerical propagation of the DHW equations. Would a multislice-calculation also include intra-band scattering?”*

Yes. Multislice simulations would include intra-band scattering. It is also possible to perform Howie-Basinski-style simulations, which include intra-band scattering, as they do not require the column approximation in contrast to Darwin-Howie-Whelan simulations. However, such multislice and Howie-Basinski simulations require a sufficiently dense spatial grid to accurately model the intra-band scattering effects (due to the stiffness of the differential equations), thus will require much higher computational efforts. Furthermore, it is not clear whether the column approximation is the limiting factor for accuracy of the calculations. For this reason, we choose the far less computational demanding and thus less time consuming Darwin-Howie-Whelan simulations.

Remark 1.3: *“Can the authors comment on the expected influence of using anisotropic elasticity theory to calculate the displacement fields?”*

For the presented case of dislocation A in GaN (which has its dislocation line along the c-axis) no change is expected, since the Burgers vector is in the elastically isotropic basal plane.

For higher symmetry cases also analytical solutions to the anisotropic theory exists (J. W. Steeds, Introduction to anisotropic elasticity theory of dislocations. Clarendon Press, Oxford, 1973). However, the dislocation line of dislocation B is not along a symmetry axis of the material, so the equations of Steeds are not applicable.

Anisotropic elasticity theory will change the details of the strain field in general less than a change of the Burgers vector direction, thus we assume that the dislocation type can still be truthfully recovered by the isotropic theory at least in the presented GaN case.

Remark 1.4: “Typos: Line 309 ‘minima’ should be ‘minimum’; Lines 891 and 900 ‘intensities intensities’”.

These have been fixed.

Reviewer #2

“In this paper the authors present a quantitative method for determining the type, depth, and dislocation line in 3 dimensions from a single 4D-STEM scan along a single projection. The quantitative work is carefully done, and the result is a novel take on a method for 3 dimensional strain determination from 4D-STEM.”

Remark 2.1: *“The precision of the dislocation depth measurement is stated to be 5nm based on the localization of a global minimum of a summed squared differences metric. However, many local minima exist, as noted by the authors. Can the authors provide a more robust statistical error bar on the depth measurement based on the quantitative squared differences metric for the confidence in the global minimum vs other of the many local minima?”*

There is no easy answer to this question. There are three major contribution to the squared differences (or mean squared errors, MSE): variations within the model, statistical variations due to noise, and due to inaccuracy of the simulation. As the experimental and calculated intensities are proportional to the overall electron dose (in the evaluated region), the MSE contributions from model variations and variations due to accuracy will scale with the squared electron dose. The statistical variations might have other dependencies to the electron dose, e. g. linear electron dose (instead of squared dose) for shot noise.

Variations within the model are the contribution that give the multi-minima landscape. In the ideal case (no noise, exact model) the MSE will evaluate to zero at the global minimum. Using the minimization of the MSE as an estimator for the model parameters (as we do in our method) requires this global minimum to be unique and truthful (for the estimator to be unbiased). We have tested this estimator property, by calculating the MSE maps between a calculated (q,x)-dataset for a given depth and thickness with the calculate datasets for all other depth and thicknesses. For each given depth and thickness the global minimum was reliably found at the given parameters and with a sufficient huge MSE difference to the second lowest minimum.

As example the left panel of the figure below shows the MSE map comparing the calculated datasets for a depth of $d=55$ nm and thickness of $t=132$ nm (same parameters as determined for dislocation A in the manuscript) with all other calculated datasets. This MSE map has been rescaled to the squared dose observed in the experiment, in order to make the MSE values itself comparable. The right panel in the figure below shows the MSE map, which is obtained by comparing the calculated and experimental (q,x)-datasets. The latter map is the same as presented in Fig. 2c of the manuscript (repeated here for convenience). The MSE of the model variation (left panel) at the optimum is obviously zero, however the overall map compares very well with the MSE map obtained from the experiment (right panel). This indicates that the variations in the “experimental” MSE are mainly caused by the sensitivity of the MSE to the model parameters and thus the model parameters (depth, thickness, ...) are robustly estimated by the MSE minimization.

The contribution of statistical variations allows to give the MSE values itself a statistical meaning. However, this requires that the contributions due to inaccuracies in the model are negligible. Then for Gaussian noise the minimization of the MSE becomes a maximum-likelihood estimator, and the MSE value in the minimum itself is chi-square distributed with a known expectation value (given by the noise variance). From this in turn in principle confidence intervals for the model parameters can be obtained from the 2nd derivatives of the MSE with respect to the parameters (or in general the Cramer-Rao lower bound resp. Fisher information matrix can be obtained).

However, in the present case the contributions of statistical variations caused by noise to the MSE is negligible compared to the contribution due to the limited accuracy of the simulation. This we mostly conclude from the fact, that no large changes of the MSE values in the global minimum are observed, when the size of the averaged area is reduced (see remark 2.2). For a shot noise-dominated MSE, it is expected that the MSE scales inversely proportionally with the “y”-extent of the spatially averaged area as the dose scales linearly with the size.

The third contribution to the MSE are inaccuracies in the simulation (which we knowingly accept here mainly for numerical efficiency). We must conclude that in the presented examples these are the dominant contributions within the global minimum. These inaccuracies are described in the discussion section of the manuscript together with mitigation strategies (e. g. to use zero-loss filtering for exclusion of inelastically scattered electrons in the experiment). We also added two sentences to the discussion, which give more detail about the role of inelastically scattered electrons, as we assume these may be limiting for thicker specimens:

For thicker specimen also inelastically scattered electrons, which experienced a plasmon loss, contribute significantly to the experimental CBED patterns [Mkhoyan2008]. These inelastically scattered electrons will result in more blurred patterns within the diffraction disk [Mendis2019] as well as a diffuse background, which, however, is mitigated by the subtraction of the empirically modeled background.

In our experience from simulations of CBED patterns for specimen thickness determination (which also suffers from the discussed inaccuracies) changing the details of the simulation, e. g. slightly changing the Debye-Waller factors, atom form factor parametrization, number of beams included in the calculation, or even changing the used simulation codes, results in an error in the determined thickness of around 5 nm (at least in the thickness range of 100 to 300 nm). For this reason, we estimated the error of our analyses in the manuscript to be around 5 nm.

As there is no straight forward statistical method to quantify this discussion, we refrain from giving such a measure and instead choose to transparently show the resulting MSE maps, which let to the parameter determination.

Remark 2.2: *“I have some significant doubts about the general applicability of this method for a wide variety of systems. The demonstration case shown involves a highly perfect crystal with only a few dislocations visible in a field of view of a few hundred nanometers. While the approximately 5nm depth resolution appears to have been fairly convincingly demonstrated for this system, the authors claim a fairly wide applicability to metals and other systems. However, most other systems will exhibit much higher dislocation and other defect densities (sometimes by orders of magnitude) than epitaxially grown semiconductors.*

For example, the q-x plots are evaluated over a range of 80 nm. If the nearest neighbor dislocation were less than 40 nm away on either side, how much would this affect the precision of the result? It also appears that diffraction patterns were integrated over a ~40nm window in the y-direction, which will have similar implications about the maximum dislocation densities that can be studied. What is the depth precision if the width of the q-x window is reduced to, for example, 20nm instead of 80 nm? What about a 10nm window? What if there is more than one dislocation (or another defect) present along the beam axis?”

We added a section to the supplementary information, where we investigated the effect of window size on the resolved dislocation depths and specimen thicknesses (Supplementary Data 3). The same global minimum was obtained and sufficiently robustly discriminated from other minima for all window sizes, even for a 11 nm window. Obviously, the required minimal window width will always vary with material, chosen systematic row, and specimen thickness. Nevertheless, much smaller windows than 80 nm are possible with the presented method.

Shrinking the area perpendicular to the x-direction (i. e. in the y-direction) will change the noise of the experimental (q,x)-plane due to shot noise. Depending on the specimen this might be mitigated by higher electron doses. In the case of dislocation A we observe only a minor change in the MSE of the global minimum (as the MSE is not noise dominated): instead of a MSE of 616 at the global minimum for the full area, the minimal MSE increases to 712 if only a quarter of the y-extent is chosen. No significant change of the MSE difference between the best and second to best minimum is observed.

We also added another section to the supplementary information, where we applied the described 3D classification method to a mechanically deformed Aluminum sample (Supplementary Data 1). With the exception of the mid-plane-ambiguity described above, also for the Aluminum sample the dislocation type and depth could be successfully identified.

In reaction to the comment of the reviewer, regarding the situation with more than one defect existing along the beam direction, we also tested our method with two dislocations being present: In the figure below a MSE map of the model variations for the geometry similar to dislocation A is shown, however including two dislocations instead of one (thickness 132 nm, both dislocations have a Burgers-vector of [000-1]). We compared the simulated (q,x)-plane for one dislocation located at a depth of 60 nm and the other one at a depth of 100 nm with simulations where the depth of both dislocations were varied. The resulting MSE map is shown below (like in the figure above, the MSE values have been scaled to the experimental electron doses of dislocation A). Along the axes the depths of both dislocations are mapped. This map obviously is symmetric around the diagonal, since both dislocations are identical. The global minimum is clearly located at the true

depths of the dislocations (please see answer 4.9 for some more remarks regarding possible regularization strategies).

Ultimately, there will be limits with respect to defect densities, when the presented method will stop to work. However, the same is true for all other methods, which recover the 3D dislocation network geometry. For instance, weak-beam dark field (WBDF) tomography also requires sufficient high levels of crystallinity and will also fail when the dislocations are too close together to discern them in WBDF images or when overlapping with contrasts of other defects.

Remark 2.3: “What is the minimum sample thickness for this method to work reliably? For example, would it be applicable to nanoparticle quantum dots (or other nanoparticle systems) in the <20nm range?”

We assume the minimum applicable thickness to be around 30 nm for the (0002) systematic row of GaN, since at this specimen thickness the first distinct features appear in the diffraction disks. At lower thicknesses the disks exhibit mainly a homogeneous intensity. However, so far we haven’t done any experiments at this thin thicknesses, as our main focus lays on thicker specimen and dislocation networks.

“Overall, I believe that this work is of high quality, and suitable for publication with a more quantitative representation of the depth precision. However, I believe that concerns about the general applicability of this method to systems with even slightly higher defect densities (which will encompass most real materials systems), suggest that eventual publication should take place in a more specialized journal than Nature Communications.”

Reviewer #3

“Niermann et al. classify and determine the 3D position of dislocations on the nanoscale using four-dimensional scanning transmission electron microscopy (4D-STEM) and electron microscopy simulations. Tomographic data acquisition is much more tedious than single projection imaging, both experimentally and in post-processing (each projection needs to be aligned). Although dislocations have been determined in three dimensions using ADF STEM [1] and electron ptychography [2] at the nanoscale without tilting, the noteworthy improvement of the work of Niermann et al. is the applicability of this method to thick STEM samples (>100nm). By acquiring 4D-STEM data over different dislocations, plotting diffraction-real-space maps (q-x maps) and comparing to multi-beam calculations via summed squared difference maps, the z-position of two different dislocations are determined in a GaN sample on a sapphire substrate. “

We thank the reviewer for bringing these references into our attention. We added a paragraph to the introduction of our manuscript, describing these techniques:

In thin GaN-samples of below 20 nm thickness, the classification and depth determination of dislocations was successfully demonstrated using multislice-ptychography [Gilgenbach2023] and depth sectioning [Yang2015]. However, in thicker specimen dynamical diffraction effects will cause an increasing challenge for these methods.

“I believe this work is well constructed and self-contained, and will be a welcome addition to the electron microscopy literature. Sample thickness limitations are one of the greatest challenges in transmission electron microscopy, and this technique accounts for this quite well. However, I am doubtful of the universality of this technique. Extensive calculations are required to find the local minimum for the sample thickness and depth of dislocation. These simulations may take considerable time, and would need to be performed for every material to be studied. If these simulations are easily implemented over reasonable time scales and, additionally, applicable to many crystal types, this would be worthy of publication in Nature Communications. For now, it seems like extensive work needs to be done to get to this point. In conclusion, I would not recommend publication in Nature Communications, but believe that a revised draft of this work would certainly be worthy of publication in Scientific Reports or a Microscopy Journal.”

“I have provided some specific feedback and comments below.”

“[1] Yang, H., Lozano, J. G., Pennycook, T. J., Jones, L., Hirsch, P. B., & Nellist, P. D. (2015). Imaging screw dislocations at atomic resolution by aberration-corrected electron optical sectioning. Nature Communications, 6(1), 7266.

[2] Gilgenbach, C., Chen, X., Xu, M., & LeBeau, J. (2023). Three-dimensional Analysis of Nanoscale Dislocation Loops with Multislice Electron Ptychography. Microscopy and Microanalysis, Volume 29, Issue Supplement_1, Pages 286–287.”

“Comments”

Remark 3.1: *“I believe Figure 1 should include a pipeline of the analytical process. For example, use text to emphasize which directions are being averaged. Then in Fig 1d insert a subfigure with the q-x map determined from Fig 1a. In its current form, it is difficult to interpret the scientific workflow from Figures 1-3 and the supporting text.”*

A sketch of the evaluation procedure has been added as panel to Figure 1 and is described in its caption. Furthermore, the description of the evaluation process is now more detailed in the results section:

As sketched in Fig. 1a and further elaborated in the methods section, a two dimensional (q,x)-plane was obtained from the 4D-STEM data. Please note, that the systematic row and thus the reciprocal space direction q can be chosen independently from the spatial x direction. For dislocation A the (q,x)-plane is obtained in the spatial dimension x along the red arrow in Fig. 1b and in the diffraction dimension q along the (0002)-systematical row (red arrow in Fig. 1c). For this plane the spatial dimension x is roughly oriented along the [2 -1 -1 0]-direction, i. e. perpendicular to the line-vector of the dislocation, with its origin x=0 nm at the intersection with the dislocation line. Its reciprocal space dimension q corresponds to the diffraction vector along the (0002)-systematical row. A similar (q,x)-plane was obtained for dislocation B.

Also the description within the method section is now more detailed and refers to the new Fig. 1a:

For further analysis the information in the region of interest of the 4D-STEM dataset was reduced to a two dimensional (q,x)-plane, which has a spatial x -dimension and reciprocal space q -dimension. This reduction of the dataset is sketched in Fig. 1a. For the dataset evaluated for dislocation A, the corresponding directions are shown Fig. 1b and 1c: in the spatial dimensions all points of the 4D dataset with scan coordinates within the sub-region marked by the red rectangle in Fig. 1b are averaged in the direction perpendicular to the line indicated by the red arrow. In the diffraction dimension all points with reciprocal space coordinates within the sub-region marked by the red rectangle in Fig. 1c are averaged in the direction perpendicular to the line indicated by the red arrow. For the dataset in (2 -1 -1 0) systematic row condition the corresponding direction of the investigated (q,x)-plane are shown in Fig. 1d and 1e.

...

Remark 3.2: *“Can you comment on the 3D precision/accuracy of this technique? The authors mention that they can identify the depth location within the limits of their simulation step sizes. Was this step size chosen because of an existing limit, or could you have used a smaller step size and located the dislocation more accurately?”*

In our experience from simulations of CBED patterns for specimen thickness determination changing the details of the simulation, e. g. slightly changing the Debye-Waller factors, atom form factor parametrization, number of beams included in the calculation, or even changing the used simulation codes, results in an error in the determined thickness of around 5 nm for thicknesses in the range of 100 to 300 nm.

In principle we could also do the simulation with smaller step sizes and thus better precision in determination of the depth. However, we think this higher precision is misleading as it neglects the inaccuracies of the simulation, in other words the accuracy is not improved by a finer step size (also see remark 2.1).

Remark 3.3: *“The advantage of this technique over ADF optical sectioning and ptychography is the relaxed sample thickness limitations. What is the maximum thickness that enables faithful depth determination? Also, what is the minimum thickness?”*

Within the revised discussion we mention limitations for thicker specimens: firstly finer CBED patterns, which are not well transferred by the detector due to its modulation transfer function, and secondly increased contributions from inelastically scattered electrons. Two sentences were added to the revised discussion to give more detail to the second point:

For thicker specimen also inelastically scattered electrons, which experienced a plasmon loss, contribute significantly to the experimental CBED patterns [Mkhoyan2008]. These inelastically scattered electrons will result in more blurred patterns within the diffraction disk [Mendis2019] as well as a diffuse background, which, however, is mitigated by the subtraction of the empirically modeled background.

Both limitations are ultimately material dependent and even can be pushed further with sufficient instrumentation, as the inelastic electrons can be removed using energy filters, and detectors with a larger field of view or a better modulation transfer function will allow for finer CBED details. If we use the empirical criterion, that the MSE of the second best minimum should be at least 10% higher than the MSE of the global minimum, we achieve an maximum applicable thickness of around 300

nm for our instrumentation, for an acceleration voltage of 300kV, and for the presented examples within the main text (GaN) and within the supplementary information (Al).

We assume the minimum applicable thickness to be around 30 nm for the (0002) systematic row of GaN, since at this specimen thickness the first distinct features appear in the diffraction disks. At lower thicknesses the disks exhibit mainly a homogeneous intensity. However, so far we haven't done any experiments at this thin thicknesses, as our main focus lays on thicker specimen and dislocation networks.

Remark 3.4: *“Some of the figure labels are incorrect (figure 2 instead of figure 1)”*

These have been fixed.

Remark 3.5: *“I believe the time taken to complete the simulations will be a critical factor in determining whether this approach will be implemented throughout the microscopy community. How long did the simulations take to run?”*

The computational demand is surprisingly low. We performed the calculation using a desktop workstation equipped with a single AMD Ryzen 9 3900X 12-core processor and 32 GB of RAM. Calculating the dataset required for the depth determination of dislocation A (for a single Burgers vector) required 812 seconds, and the calculation for the dataset for a single Burgers vector for the newly added Aluminum dislocation in the supplementary information required 913 seconds.

The huge advantage of Darwin-Howie-Whelan calculations in the column approximation is that the spatial grid is decoupled from the equations. This is a difference to e. g. multislice calculations or Howie-Basinski-style calculations, where a sufficient dense sampling of the spatial grid is required in order to get accurate results.

We added a section on the computational demands to the supplementary information (Supplementary Data 4), to demonstrate the numerical efficiency of the presented methods.

Remark 3.6: *“I found the supplementary movies to be very helpful in understanding the findings. I suggest explicitly referring to these movies (i.e.. Movie S1, S2, etc) in a revised draft.”*

We strengthened the reference to this movies in the revised draft by making more clear to which movie we refer at the corresponding points in the manuscript (cf. Remark 3.7).

Remark 3.7: *“From the squared difference maps in Fig. 2c and 3c it can be seen that several local minima exists. However, the global minima always was sufficient well identified. This was confirmed by visual inspection of the other possible candidates.”*

“Could this ‘visual inspection’ step be replaced with a quantitative comparison metric?”

We removed sentences referring to visual inspection as they indeed may be misleading. We actually meant that beside the quantitative criterion (i. e. the MSE of the global minimum being significantly lower than the MSE of the other minima) we additionally manually verified that the simulated (q,x)-planes of the other candidates were visually very different.

Within the revised manuscript, we now only refer to the differences in the MSE. For the dislocation A case:

The minimum for the matching parameters is quite distinct: the MSE of the second lowest minimum was 42% larger than the MSE of the global minimum. The general dissimilarities of the simulated (q,x)-planes for different parameters of depth and thickness, can be seen from Supplementary Movies 1 and 2.

for the dislocation B case:

The map of the MSE between experimental and calculated (q,x)-planes in Fig. 3c shows that the minimum is not as distinct as in the case for dislocation A: the MSE of the second lowest minimum was 18% larger than the MSE of the global minimum (see also the parameter sweep in the Supplementary Movies 3 and 4).

and removed the mentioned sentence completely from the discussion.

Reviewer #4

“In their paper titled "Revealing the Three-Dimensional Location and Classification of Dislocations from Single 4D-STEM Measurements," the authors present a new method for determining the three-dimensional location and type of dislocations in materials using single 4D-STEM measurements. Dislocations play a crucial role in governing material properties, and knowledge of their three-dimensional topology is essential for material engineering. While two-dimensional projections of dislocation networks can be obtained using conventional transmission electron microscopy (TEM) images, reconstructing the three-dimensional structure has been challenging and requires tomographic tilt series with high angular ranges.”

“The authors propose a new approach that utilizes the strain field caused by dislocations to induce inter-band scattering between the electron's Bloch waves within the crystal. This scattering leads to characteristic interference patterns in the diffraction pattern, which can be used to identify the type and depth of the dislocations”

“They demonstrate the effectiveness of this method by applying it to heteroepitaxial films of wurtzite-type GaN grown on a sapphire substrate. The proposed method involves capturing zeroth order Laue zone (ZOLZ) convergent beam electron diffraction (CBED) patterns using 4D-STEM. The inhomogeneities caused by dislocations alter electron propagation, creating characteristic interference patterns. Multi-beam calculations were crucial, enabling the comparison of simulated CBED patterns with experimental data, ultimately determining dislocation types and 3D positions. By comparing experimental and simulated data in the (q, x)-plane, the authors validated their method, showcasing its accuracy.”

“I agree with the authors that the technique has an interesting potential for beam sensitive materials or during in situ (cooling-heating for instance) experiments requiring fast measurements. For this reason I am happy to recommend this work for publication in Nature Communications once the comments below are considered.”

“General comments; Clarity of presentation:”

“The paper contains highly technical language and complex methodologies and could benefit from clearer explanations (and/or additional references) for readers who are not experts in the field. Clearer explanations, possibly accompanied by additional references, are essential, especially considering the diverse readership of Nature Communications, spanning various fields. Specific sections of the methods, in particular, are difficult to understand (in my opinion)”

Remark 4.1: “L 570-571: ‘The propagation is performed using a 4th-order Runge-Kutta scheme with a step size of 0.1 nm.’. Could the authors elaborate or at least provide a reference?”

We added a reference to the “Numerical Recipes”-Book in the Methods section of the main text.

Remark 4.2: “L 673-675: ‘The other parameters were minimized for a given set of (t, d) using a quasi-Newton method with derivatives estimated from 2-point finite differences.’ Could the authors elaborate on the implementation of this minimization?”

We added a reference to the numerical Python package “scipy” and references to the used method itself (instead of referring to a quasi-Newton method) in the Methods section of the main text.:

The other parameters were numerically minimized for a given set of (t,d) using the Broyden-Fletcher-Goldfarb-Shanno method implemented in the numerical *Python* package *scipy* [scipy, Nocedal2006].

“In addition, some figures could be added to the manuscript or at least to the extended data to help the reader:”

Remark 4.3: L 225-231: ‘For positions sufficient far away from the dislocation the pattern resembles the CBED pattern of an unstrained crystal’ Could the author provide a CBED pattern from an unstrained region of the crystal (either from their own experimental data or at least a reference to a work showing CBED patterns from unstrained crystals).

We added simulated CBED patterns of unstrained GaN for the given thickness and for matching specimen tilts for the (0002) systematic row as Supplementary Figure 5.

Remark 4.4: L309-313: ‘We attribute this to the weaker scattering within (2-1-10)-systematic row compared to (0002) and the thicker specimen, the former leads to less distinct CBED patterns, while the latter leads to finer CBED patterns.’ Given the fact the authors are discussing the differences the differences between the CBED patterns within the (2-1-10) and (0002) systematic rows, I think it would be helpful to show a diffraction pattern averaged over all scan coordinates within the red rectangle of Fig 2.b.”

Fig. 1 now includes the diffraction patterns averaged for all scan points within the red rectangles for both dislocations. Accordingly, the diffraction pattern for dislocation B within (2 -1 -1 0) systematic row was removed from the supplementary information as it is now moved to the main text.

Remark 4.5: “Fig. 2c and 2d. In addition to the sum squared difference maps, incorporating a line profile passing through the global minima would enhance the visualization”

Such profiles are added as Supplementary Figure 4.

“Comparison with existing literature and discussion.”

Remark 4.6: “The paper provides a comprehensive review of the existing literature on the investigation of dislocation networks using tomographic methods. The presentation of the limitations of current techniques, such as X-ray topography tomography and tilt-tomography TEM, and the experimental challenges and constraints associated with these methods is quite convincing. However, some additional references could be useful for the reader”

“1) Williams, D.B. and Carter, C.B. (2009). *Transmission Electron Microscopy: A Textbook for*

Materials Science. Can be used to support the discussion on the use of 4D-STEM and the analysis of CBED patterns”

“2) Hull, D. and Bacon, D.J. (2011). Introduction to Dislocations. Can provide additional background information on the classification and characterization of dislocations and be used to introduce the $g \cdot b = 0$ criterion together with ref [8]”

“3) Sutton, A.P. and Balluffi, R.W. (1995). Interfaces in Crystalline Materials or another reference dealing with interfaces in crystalline materials which can be related to the discussion on the interaction between dislocations and the strain field in the crystal.”

We agree that these references to these books are helpful for the reader. References to these books have been added at appropriate places in the main text.

Remark 4.7: *“The paper would also benefit from some contextualization explaining the choice of the GaN/Sapphire system. Is it a model system? How dislocations can affect its properties, real world applications...?”*

This system was mainly chosen for the purely practical reason: it is the system with the most dislocations in our daily work, which are mainly on epitaxial semiconductor materials. A reference on the impact of dislocations on the opto-electrical property of GaN devices is given in the introduction.

Remark 4.8: *“Conclusion. The conclusion is concise and clear, but you could emphasize the practical implications of the findings more explicitly. How can this method benefit material science and engineering? The conclusion could be more explicit about the significance of the findings. Summarize the key discoveries and their implications. Additionally, discuss potential future directions for research based on the current findings. I am also surprised that the authors didn't mention the potential applicability of Machine Learning approaches in this context Utilizing convolutional neural networks trained on a dislocation fingerprint database could potentially enhance the method's automation.”*

We added sentences to emphasize this implications more explicitly:

... From the 3D structure of dislocation networks more information about crystal plasticity and the spatial interactions of dislocation with interfaces may be obtained. ...

... For instance, future in-situ experiments may study the evolution of a dislocation network during annealing or under external load. ...

We also thank the reviewer for bringing this potential application of machine learning techniques into our attention and added a sentence mentioning this approach:

Such an automation might also be a potential application for machine learning approaches, where neural networks have been trained on fingerprint databases.

“Additional questions:”

Remark 4.9: *“L 249-253 ‘Smaller deviations are mainly found in the upper region with $x < -25$ nm and are probably caused by the strain field of dislocation D.’ Is it possible to take into account interactions between dislocations in the model? Would it be challenging to implement?”*

Within the Darwin-Howie-Whelan simulation only a formulation of the displacement field is needed, so far the analytical formulas found in the book of Hirth & Lothe were used for this. Within the limitations of linear elasticities the displacements of the dislocations add up, and implementing a second dislocation is straight forward.

However, when the depth and type of a second dislocation are not known this will significantly increase the search space and will have a significant impact on the overall simulation time (if these parameters are known for the additional dislocation, the simulation time remains nearly the same).

We tested the situation, where two dislocations are present along the beam, the global MSE minimum still correctly identifies the true depths (see answer 2.2).

However, for two dislocations we see promising possibilities for regularization of the MSE estimators from prior knowledge: if the course of the two dislocation lines are different over the overall 4D-STEM datasets one could use this to regularize their depths at the crossing points of the lines with the depth of the nearby regions, where only a single dislocation is found. Furthermore, one could use the elastic energy between both dislocations as a regularization and penalize geometries which are energetically unfavorable within the MSE minimization.

Remark 4.10: *“It is also mentioned that line vector directions were taken from the corresponding dark field images. Did they authors try to vary slightly the line direction in the simulations to evaluate the effect of this parameter on the calculations?”*

We tried it, but the overall effect is negligible, at least for our cases. We attribute this to the fact, that our evaluation procedure is basically a projection (onto the q - and x -axis). In the present geometries a rotation of the line causes the projection of the displacement fields (in size and location) to vary only slightly, as their projection changes with the cosine of the rotation angle.

Remark 4.11: *“Typos: L 210-211 “along the red arrow in Fig. 2a in the spatial dimension x and along the systematical row in the diffraction dimension q (red arrow in Fig. 2c). I think the authors were referring to Fig. 1a and 1c, respectively”*

The references to the figures have been fixed.

REVIEWERS' COMMENTS

Reviewer #1 (Remarks to the Author):

All comments and questions were answered satisfactorily. The paper should be published as is.

Reviewer #2 (Remarks to the Author):

The authors have satisfied my concerns through their revised submission, and I now recommend the work for publication.

I very much appreciate the additional data presented for a deformed Aluminum system, which goes a long way toward demonstrating the versatility of their new technique. It is also highly encouraging to see similar dislocation thickness results for much smaller sizes of integration windows. I am much more convinced now that the technique may have applications beyond the initial case presented, which was a bit niche on its own.

As a result of the revision, there now appear to be two sets of supplementary figures with overlapping numbers. That should be addressed before publication.

Reviewer #3 (Remarks to the Author):

Niermann et al. have addressed my comments from the previous manuscript draft. The figures are described more clearly, and the applications/limitations of the technique have been explicitly stated. The addition of a second materials system in the supplementary material, as well as the modest computational times (which were my primary concerns with the previous manuscript draft) demonstrates generality of the technique. Therefore, I recommend publication of this revised manuscript in Nature Communications.

Reviewer #4 (Remarks to the Author):

I have reviewed the revised manuscript, and I'm pleased with the authors' comprehensive response to my previous comments. Their revisions significantly strengthen the manuscript, addressing concerns and enhancing clarity. I recommend this work for publication in Nature Communications.

Response Letter

First, we thank the reviewers for taking the time and effort to provide constructive comments, which have been very helpful in making improvements to the manuscript.

Response to the reviewers

Reviewer #2 commented:

As a result of the revision, there now appear to be two sets of supplementary figures with overlapping numbers. That should be addressed before publication.

We addressed this issue and all figures within the Supplementary Information now use a consistent numbering scheme.

Additional changes

- The experimental parameters of the Aluminum sample (Supplementary Note 1) are now stated more precisely: compared to the measurements presented in the main text the dwell time was increased to 50 ms, while the beam current was roughly reduced by a factor of 8. There were no special reasoning for choosing these parameters. As discussed in the previous response letter, we consider all presented measurements to be not limited by noise nor the specimen to be dose limited in any way, thus we expect the actual electron dose (rate) to be insignificant for the presented findings.
- The crystal indices in Supplementary Fig. 1b and 1c were erroneously given in the wrong direction. This is now fixed in accordance to the text and other figures.
- All other changes to the manuscript were needed to comply with the formatting guide. Beside small formatting changes this required the rearrangement/reformulation of two paragraphs of the introduction (see highlighted difference below).
- Beside the two points given above, all changes of the supplementary information were small formatting changes needed to comply with the formatting guide.

TITLE

Three dimensional classification of dislocations from single projections

AUTHOR LIST

Tore Niermann^{1*}, Laura Niermann¹, Michael Lehmann¹

AFFILIATIONS

¹ Technische Universität Berlin, Institut für Optik und Atomare Physik, Straße des
17. Juni 135, 10623 Berlin, Germany

These authors contributed equally: Tore Niermann, Laura Niermann

Correspondence should be addressed to T.N. (email: tore.niermann@tu-berlin.de)

ABSTRACT

Many material properties are governed by dislocations and their interactions. The reconstruction of the three-dimensional structure of ~~the~~ a dislocation network so far is mainly achieved by tomographic tilt series with high angular ranges, which is experimentally challenging and additionally puts constraints on possible specimen geometries. Here, we show a ~~new~~ way to reveal the three dimensional location of dislocations and simultaneously classify their type from single 4D scanning transmission electron microscopy measurements. The dislocation's strain field causes inter-band scattering between the electron's Bloch waves within the crystal. This scattering in turn ~~causes results in~~ characteristic interference patterns with sufficient information to identify the dislocations type and depth in beam direction by comparison with multi-beam calculations. We expect the presented measurement principle will lead to fully automated methods for reconstruction of the three dimensional strain fields from such measurements with a wide range of applications in material and physical sciences and engineering.

INTRODUCTION

Dislocations and their interaction are responsible for a wide range of material properties, ranging from strengthening of metals and alloys [1] to efficiency in semiconductor laser devices [2]. Thus, knowledge of the three dimensional topology of dislocation networks is crucial for material and interface engineering [3]. A two-dimensional projection of dislocation networks can be readily obtained by conventional (scanning-) transmission electron microscopy (S/TEM) images [4].

The three dimensional topology of dislocation networks is currently mainly investigated by means of tomographic methods. The first reconstruction was done using X-ray topography tomography, however at low resolution [5]. Tilt-tomography of weak beam dark field images within the TEM allows the investigation of dislocation networks at 5 nm resolution [6]. By a combination of tilt-tomography STEM with Fourier filtering even higher resolutions might be achievable [7], even though the resolution limits of such an approach are debated [8]. These tomographic techniques have in common that a large number of images must be obtained with high tilt ranges (typically 25 to 200 micrographs with angular ranges > 120 degrees), furthermore they are often troubled by the missing wedge problem and dynamical scattering effects [9].

In thin GaN-samples of below 20 nm thickness, the classification and depth determination of dislocations was successfully demonstrated using multislice-ptychography [10] and depth sectioning [11]. However, in thicker specimen dynamical diffraction effects will cause an increasing challenge for these methods.

For classification of dislocation types the projected line vector of a dislocation is readily obtained from conventional S/TEM images, and the direction of the Burgers vector \mathbf{b} can be determined by the famous $\mathbf{g} \cdot \mathbf{b} \neq 0$ criterion for reflections \mathbf{g} [1, 12, 13]. From convergent beam electron diffraction (CBED) patterns of defocused probes (convergent beam imaging) even the Burgers vector's length and sign can be determined from higher order Laue zone line splittings at the dislocation line [14].

Instead of mixing diffraction and imaging information like done in convergent beam imaging we instead use 4D-STEM [15] to collect zeroth order Laue zone (ZOLZ) CBED pattern

with non-overlapping disks for each scan point with focused probes. These measurements are performed under illumination conditions, where the resulting CBED patterns are governed by dynamical diffraction. Under such conditions the propagation of electrons through the crystal is affected by inhomogeneities $\frac{\partial}{\partial z}(\mathbf{g} \cdot \mathbf{u})$ of the displacement field \mathbf{u} in beam direction z and their depth (z -position) within the specimen [16–18]. These inhomogeneities are for instance local shears and rotations of the lattice caused by the dislocation.

The effects of these inhomogeneities on the electron beam propagation can be understood in the Bloch wave picture [16]. In an unstrained crystal, the Bloch waves propagate undisturbed through the crystal with different longitudinal wave vector components depending on the Bloch wave band. Eventually, the interference between the individual Bloch waves at the exit surface of the specimen is observed in the diffraction pattern. The different longitudinal Bloch wave vector components cause the well known beating of the intensity with crystal thickness t (Pendellösung). Inhomogeneities of the displacement field cause the electrons to scatter longitudinally between Bloch wave bands (inter-band scattering). Due to the interference of the Bloch waves at the exit surface this redistribution of Bloch wave excitations is detectable in the diffraction pattern. Also a lateral scattering of the Bloch waves on the respective dispersion surfaces of a Bloch wave band (intra-band scattering) occurs. However, this intra-band scattering has only a minor effect on the resulting diffraction patterns compared to the inter-band scattering. Since the interference of the Bloch waves at the exit surface not only depends on the difference of the longitudinal components of the Bloch wave vectors but also on the distance traveled within the crystal, the resulting interference is also sensitive on the depth of the inhomogeneity within the specimen.

A CBED pattern allows the inspection of these Bloch wave interferences for several incident beam directions at once. ~~With a 4D-STEM measurement also the spatial variations of these CBED patterns with distance from the dislocation are observable~~

Nevertheless, a fundamental ambiguity of electron scattering for centro-symmetric scattering geometries exists [22]: displacements fields, which exhibit an antisymmetric mirror symmetry with respect to the specimen midplane in beam direction (i.e. $u_z(z) = -u_z(t - z)$ where t is the specimen thickness) will result in the same diffraction pattern.

For a given Burgers vector and line vector of a dislocation the displacement field can be analytically calculated in simple cases like for isotropic elasticity [20], or numerically in general cases [21]. For a known strain field and a given electron probe position relative

to the dislocation, the expected CBED patterns can be efficiently simulated by means of multi-beam calculations [12]. ~~By comparison of calculated patterns with the CBED patterns observed experimentally,~~

~~In this work, we show a method to uniquely determine~~ the type and three dimensional position of ~~the dislocation can be uniquely~~ dislocations (with the exception ~~below~~) ~~determined as long as enough Bloch waves are excited. The latter condition is achieved by recording the CBED patterns under many-beam conditions instead of two-beam conditions.~~

~~Nevertheless, this technique can not resolve a fundamental ambiguity of electron scattering for centro-symmetric scattering geometries [22]: displacements fields, which exhibit an antisymmetric mirror symmetry with respect to the specimen midplane in beam direction (i.e. $u_z(z) = -u_z(t-z)$ where t is the specimen thickness) will result in the same diffraction pattern.~~

~~Here we use this technique to determine of the mentioned midplane symmetry).~~ For this, spatial variations of CBED patterns with distance from the dislocation are extracted from a 4D-STEM measurement and these patterns are compared to calculated patterns. We demonstrate this method by determining the depth and type of dislocations in within a hetero-epitaxial films of wurtzite-type GaN on a sapphire substrate.

RESULTS

Specimen overview

The specimen is a wurtzite-type GaN film grown in [0001]-direction on a sapphire substrate. The lattice mismatch between ~~both materials~~ the GaN layer and the substrate result in dislocations threading through the film in the growth direction [23]. Perfect dislocations ~~of within~~ this material system ~~correspond to are~~ those of the hexagonal lattice and are characterized by Burgers vectors \mathbf{b} of $\mathbf{a} = \frac{1}{3}\langle\bar{1}120\rangle$, $\mathbf{c} = \langle 0001\rangle$, or $\mathbf{a} + \mathbf{c} = \frac{1}{3}\langle\bar{1}123\rangle$, with \mathbf{a} and \mathbf{c} corresponding to the base vectors of the lattice [20, 24].

RESULTS

A region roughly 750 nm above the interface between the GaN-buffer and the sapphire substrate was investigated under two systematic row conditions, namely the (0002) and

($2\bar{1}\bar{1}0$) systematic rows only in order to demonstrate the method for different excitation conditions (see Supplementary Fig. ~~4~~5 for a larger area image overview). Annular dark field (ADF) images of the investigated region for both systematic-row conditions are shown in Fig. 1b and 1d. This region was selected since it exhibits several dislocations of different types. These dislocations are emerging threading dislocations rooted in the interfacial misfit. Within this region a dislocation (marked A in the figure) with a line vector along the $[0001]$ direction is observable, which is only strongly visible in the (0002) systematic row. The dislocations B, C, and D with line vectors roughly 45 degrees inclined to the $[0001]$ -direction are only strongly visible in the $(2\bar{1}\bar{1}0)$ systematic row. ~~Additional~~Additionally, two basal stacking faults can be seen (E). Using the $\mathbf{g} \cdot \mathbf{b}$ criterion the set of possible Burgers vectors for these dislocations can already be reduced to: $\pm[0001]$ for dislocation A, and $\pm\frac{1}{3}[2\bar{1}\bar{1}0]$, $\pm\frac{1}{3}[\bar{1}\bar{2}10]$, $\pm\frac{1}{3}[11\bar{2}0]$ for dislocations B, C, and D. Partial dislocations can be ruled out since the dislocations are not connected to other extended defects. Since the Burgers and line vectors for dislocation A are parallel this dislocation is of screw type, while dislocations B, C, and D are of mixed type. In the following we will further investigate dislocation A and B. The investigation of dislocations C and D is similar to the analysis of dislocation B. Please note, that the circular features present in the right half of the images originate from carbon contamination during the microscopy session and are not caused by crystalline defects in the image.

Study of dislocation A

As sketched in Fig. 1a and further elaborated in the ~~methods~~Methods section, a two dimensional (q, x) -plane was obtained from the 4D-STEM data. Please note, that the systematic row and thus the reciprocal space direction q can be chosen independently from the spatial x direction. For dislocation A the (q, x) -plane is obtained in the spatial dimension x along the red arrow in Fig. 1b and in the diffraction dimension q along the (0002) -systematical row (red arrow in Fig. 1c). For this plane the spatial dimension x is roughly oriented along the $[2\bar{1}\bar{1}0]$ -direction, i.e. perpendicular to the line-vector of the dislocation, with its origin $x = 0$ nm at the intersection with the dislocation line. Its reciprocal space dimension q corresponds to the diffraction vector along the (0002) -systematic row. A similar (q, x) -plane was obtained for dislocation B.

The resulting (q, x) -plane for dislocation A is shown in Fig. 2a. Along the q -direction the CBED patterns of the 5 innermost reflections of the systematic row are clearly visible as separated intervals. Along the x -direction the variation of these CBED patterns in dependence of the distance x to the dislocation can be seen. For positions sufficient far away from the dislocation the pattern resembles the CBED pattern of an unstrained crystal (see Supplementary Figure 5 Fig. 9). Such a behavior is seen in Fig. 2a, where the patterns further away ($|x| \gtrsim 35$ nm) from the dislocation become constant with x . The difference between the patterns for $x \lesssim -35$ nm and $x \gtrsim 35$ nm can be explained by a bending of the specimen caused by the far field of the dislocation's strain field. Closer to the dislocations core ($|x| \lesssim 20$ nm) more complicated features are observed, which are caused by the stronger strain in the near field of the dislocation. At the dislocation core itself a discontinuity of the patterns is observable.

Fig. 2b shows the calculated (q, x) -plane for the parameters best matching this experimental dataset. Details on the calculation can be found in the methods-Methods section. A very good agreement between the experimental and simulated (q, x) -planes can be found. Typical CBED features like the periodic fringes within each reflection occur at similar points and with similar intensities. Smaller deviations are mainly found in the upper region with $x < -25$ nm and are probably caused by the strain field of dislocation D. Also minor deviations are found at the projected dislocation core. However, these are expected due to inaccuracies of the simulation at the core (see methods-Methods section). The best match was found for a specimen thickness of $t = 132$ nm, a depth of the dislocation core of $d = 55$ nm, an incident beam tilt of $\tau = 3.7$ mrad, a Burgers vector of $\mathbf{b} = [000\bar{1}]$, and line vector of $[0001]$. A Burgers or line vector of opposite sign would result in a (q, x) -plane with a flipped x -direction.

More insight in the quality of the match can be gained from the mean squared error (MSE), i.e. the mean squared intensity difference between experiment and calculation, which is shown in Fig. 2c for different specimen thicknesses t and depths of the dislocation d . The minimum for the matching parameters is quite distinct: the MSE of the second lowest minimum was 42% larger than the MSE of the global minimum. The general dissimilarities of the simulated (q, x) -planes for different parameters of depth and thickness, can be seen from Supplementary Movies 1 and 2.

Study of dislocation B

Fig. 3a shows the (q, x) -plane of dislocation B obtained from the dataset in $(2\bar{1}\bar{1}0)$ systematic row. The x -direction of this plane is indicated by the arrow in Fig. 1d and its q -direction is oriented along the systematic row in diffraction space (see Fig. 1e). The best matching calculation was found for a specimen thickness of $t = 172$ nm, a depth of the dislocation core of $d = 85$ nm, an incident beam tilt of $\tau = 0.83$ mrad, a Burgers vector of $\mathbf{b} = \frac{1}{3}[\bar{1}2\bar{1}0]$, and line vector parallel to $[14\bar{7}\bar{7}15]$. The simulated (q, x) -plane for these parameters is shown in Fig. 3b. While experiment and calculation in generally match well, some differences especially close to the core ($|x| \lesssim 5$ nm) in the (0000)-beam can be found, which we attribute to the inaccurate simulation of the effects of the strong strain field close to the core. The map of the MSE between experimental and calculated (q, x) -planes in Fig. 3c shows that the minimum is not as distinct as in the case for dislocation A: the MSE of the second lowest minimum was 18% larger than the MSE of the global minimum (see also the parameter sweep in the Supplementary Movies 3 and 4).

A comparison of the experimental (q, x) -plane with calculations for a Burgers vector of $\mathbf{b} = \frac{1}{3}[2\bar{1}\bar{1}0]$ shows significant differences (see Supplementary Figure 2 Fig. 6). However, the comparison with the calculations for an Burgers vector of $\mathbf{b} = \frac{1}{3}[11\bar{2}0]$ shows a similar matching calculation for a dislocation depth of $d = 90$ nm (see Supplementary Figure 3 Fig. 7). This similarity corresponds to the aforementioned mid-plane ambiguity of electron diffraction, since the $(2\bar{1}\bar{1}0)$ -systematic row is along a centro-symmetric direction, the directions $[\bar{1}2\bar{1}0]$ and $[11\bar{2}0]$ only have an opposing component in beam direction, thus result in a displacement field with flipped components in beam direction, and the depth of both dislocation core are approximately located at similar distances but in opposing directions from the midplane. Since the centro-symmetry is broken in the $[0001]$ direction, no such mid-plane ambiguity exists for dislocation A.

DISCUSSION

From the MSE maps in Fig. 2c and 3c it can be seen that several local minima exists. However, the global minimum was always sufficient well identified. The reported thicknesses were verified by electron holography [25] as an alternative method for thickness determina-

tion (see Supplementary DataNote 2). Within Supplementary DataNote 1 we additionally demonstrated the described technique on a mechanically deformed Aluminum sample as alternative material system, where dislocation depth and type could also be successfully identified. The described method even successfully identifies the depth although with a less distinct minimum, when the spatial extents of the (q, x) -planes in the present example is reduced to as low as 11 nm, compared to the 89 nm in Fig. 2 (see Supplementary DataNote 3).

In the second example above the global minima was less prominent than in the first example. We attribute this to the weaker scattering within $(2\bar{1}\bar{1}0)$ -systematic row compared to (0002) and the thicker specimen, the former leads to less distinct CBED patterns, while the latter leads to finer CBED patterns. For thicker specimen (like also observed for the Aluminum example in the Supplementary DataNote 1) also inelastically scattered electrons, which experienced a plasmon loss, contribute significantly to the experimental CBED patterns [26]. These inelastically scattered electrons will result in more blurred patterns within the diffraction disk [27] as well as a diffuse background, which, however, is mitigated by the subtraction of the empirically modeled background.

We expect that the method can be improved in future: a possible improvement might be the use of zero-loss filtered diffraction patterns, which exhibit a higher quality for quantitative comparison of the elastic signal [28]. Also the a-priori knowledge that dislocations lines either end at the surfaces or in interactions with other defects can be used in combination with tracing the dislocation line at several positions.

The ambiguity due to mid-plane symmetry can not be resolved by the presented method for centro-symmetric cases. Replacing the sample with a sample mirrored at the mid-plane, however, should make no difference for nearly all practical applications, as due to the centro-symmetry the material properties remain the same. Even in cases where the symmetry is broken by the geometry of the sample, e. g. by interfaces, this direction would not be placed in beam direction in a typical S/TEM experiment. A 3D model of dislocation networks might still be obtained by selecting consistent depths and types for neighboring regions of a dislocation, e. g. by a suitable regularization.

In the presented cases we could determine the depth of the dislocation within the step-size of 5 nm used in the calculations, the specimen thickness could be determined with a similar precision. Beside the quality of the comparison metric this precision is also determined

by the difference in longitudinal components of the Bloch wave vectors as well as number of Bloch waves excited, both depend on the material and the excitation conditions, such that even higher precisions might be possible. The accuracy of the depth also depends on the strain fields and structure factors used as input into the calculations. The isolated atom approximation and the absorptive optical potentials limit the accuracy of the multi-beam calculations. The strain fields used in the calculations assumed infinite volumes and isotropic elasticity. Nevertheless, we consider these approximations to be accurate enough for the claimed 5 nm precision. However, for dislocations close to the specimen surfaces the calculations are not accurate enough, and more complex strain models must be investigated that included for example relaxation effects.

CONCLUSION

The ability to three dimensionally locate the dislocation within a specimen, while simultaneously classify their type provides an extremely powerful way for the investigation of dislocation networks. From the 3D structure of dislocation networks more information about crystal plasticity and the spatial interactions of dislocation with interfaces may be obtained. This 3D classification is possible with a single 4D-STEM measurements within the limitations of the $\mathbf{g} \cdot \mathbf{b}$ criterion. This technique is in principle not limited to the systematic row, and might be also performed close to zone axis conditions, where more Burgers vector orientations can be covered in a single measurement.

Compared to tomographic methods a single 4D-STEM measurement might be performed much faster and with less dose, such that dislocations networks can be investigated in more beam sensitive materials and also on more dynamic conditions, e.g. during a in-situ heating/cooling experiment. For instance, future in-situ experiments may study the evolution of a dislocation network during annealing or under external load.

Furthermore, we expect this measurement technique can be developed further into an automated determination and classification scheme using a pre-calculated data base of dislocation's fingerprints and incorporation of further a-priori knowledge about the material system. Such an automation might also be a potential application for machine learning approaches, where neural networks have been trained on fingerprint databases.

METHODS

Experiment

The specimen slab was prepared with surfaces close to the $(01\bar{1}0)$ crystal planes by a conventional cross-section TEM preparation method consisting out of mechanical grinding followed by ion milling until electron transparency. For 4D-STEM measurements the specimen was rotated by roughly 4 degrees from the $[0\bar{1}10]$ zone axis into the respective systematic conditions.

For the 4D-STEM measurements the region was scanned on a 256×256 point grid with sampling steps of 0.78 nm. For each scan point the central part of the diffraction pattern was recorded with a ~~Quantum Detector MerlinEM~~ Quantum Detector MerlinEM single chip detector on a 256×256 point grid with 0.75 nm^{-1} sampling. The datasets were obtained using a ~~JEOL GrandArm F2~~ JEOL GrandArm F2 microscope operated at 300kV in Cs-corrected STEM-mode with an illumination semi-convergence angle of $\Theta = 3.3 \text{ mrad}$ and a dwell time of 1.3 ms.

Data processing

For further analysis the information in the region of interest of the 4D-STEM dataset was reduced to a two dimensional (q, x) -plane, which has a spatial x -dimension and reciprocal space q -dimension. This reduction of the dataset is sketched in Fig. 1a. For the dataset evaluated for dislocation A, the corresponding directions are shown Fig. 1b and 1c: in the spatial dimensions all points of the 4D dataset with scan coordinates within the sub-region marked by the red rectangle in Fig. 1b are averaged in the direction perpendicular to the line indicated by the red arrow. In the diffraction dimension all points with reciprocal space coordinates within the sub-region marked by the red rectangle in Fig. 1c are averaged in the direction perpendicular to the line indicated by the red arrow. For the dataset in $(2\bar{1}\bar{1}0)$ systematic row condition the corresponding direction of the investigated (q, x) -plane are shown in Fig. 1d and 1e. This data reduction to a (q, x) -plane corresponds to the common operation of obtaining one dimensional profiles from two dimensional images. However, here this operation is performed twice, once in the spatial dimensions and once in the diffraction dimensions of the 4D dataset.

For display purposes the diffraction patterns in Fig. 1c and Fig. 1e were mirrored and rotated to match the orientation of the scanning grids in Fig. 1b and 1d. The ADF images in Fig. 1 were calculated from the 4D datasets by integrating over the scattering angles in the range from 3.3 mrad to 10.2 mrad for each scan point.

Calculations

In order to attribute the Bloch wave interference patterns visible in the (q, x) -planes to specific dislocation types and dislocation depths multi-beam scattering simulations are performed, where all beams of the respective systematic rows within $\pm 30 \text{ nm}^{-1}$ were considered. These simulations are based on the numerical propagation of the Darwin-Howie-Whelan (DHW) equations along the beam direction (z -direction, here assumed parallel to the $[01\bar{1}0]$ crystal direction) and are performed within the column approximation [12, 16]. The propagation is performed using a 4th-order Runge-Kutta scheme [29] with a step size of 0.1 nm.

The Fourier coefficients of the specimen's potential (including absorption effects as optical potential) are calculated for an unstrained GaN-crystal within the isolated atom approximation from parameterized data [30]. The effect of the displacement field $\mathbf{u}(x, z)$ is modeled as additional position dependent geometric phase of these coefficients [31]. All simulations were performed with the line vector along the y -direction, such that the displacement field is constant in that direction. Within the column approximation scattering due to lateral changes of the displacement is ignored. Thus the resulting intensities only carry a parametric dependence to the lateral position x . However, the effects of displacement field inhomogeneities in the z -direction are fully included.

Even though the DHW-equations describe the dynamical diffraction in a plane-wave base their numerical propagation also can be used to correctly model the inter-band scattering of Bloch-waves. In the Bloch wave picture the column approximation corresponds to the restriction to inter-band scattering (opposed to intra-band scattering) [16]. However, following the discussion above we consider the effect of lateral scattering on the resulting diffraction patterns to be negligible except for the uttermost core of the dislocation.

For the simulations the well-known analytical displacement fields of dislocations in elastically isotropic media are used [20]. We consider the effects of the elastic anisotropy to be

negligible within the validity limits of the calculation. The direction of the line vectors of the dislocations and the possible types of Burgers vector are taken from the corresponding dark field images. Simulations in dependence of the dislocation core depth d (measured from entrance surface) were performed for all Burgers vectors compatible with the observed $\mathbf{g} \cdot \mathbf{b}$ case.

Using k to characterize the component of the incident wave vector along the systematic row, the simulation returns the diffraction intensities $I_g(x, -k; t, d)$ for all beams g included for a given specimen thickness t and dislocation depth d . The intensities $I(x, q; t, d)$ corresponding to the intensities obtained in a (q, x) -plane from the scanning convergent beam experiments are eventually given by

$$I(x, q; t, d) = N \sum_g I_g(x, q + k_0 - g; t, d) \text{ for } g \text{ with } \lambda|q + k_0 - g| < \Theta, \quad (1)$$

where N is the total intensity in the beam, k_0 is the lateral component of the central beam's wave vector along the systematic row and is used to characterize the incident beam's tilt $\tau = \lambda k_0$. Furthermore, Θ is the illumination's semi-convergence angle and λ the vacuum wave length. The additional restriction regarding g mimics the effect of the illumination aperture. To match the grid of the experimental data the simulation data was bi-linearly interpolated within the (q, x) -plane. Please note, that in all presented experiments the semi-convergence angle is smaller than the Bragg angle, such that the CBED disks of the individual diffraction do not overlap and no interference effects between the beams need to be considered.

Comparison

Beside the Bragg reflections also a diffuse background can be found within the experimental data between the reflections. This diffuse background originates from scattering at the amorphized surfaces due to specimen preparation, from carbon contamination within the microscope and from inelastic scattering. For a quantitative comparison of the experimental data with the calculations this diffuse background is empirically modeled as a broad Gaussian intensity distribution, which is added to the calculated intensities. The Gaussian is adjusted in height and width such, that it remains below the intensity minima found between the reflections in the experimental data. Eventually, the calculated intensity data

is convoluted with the point spread function of the detector [32], which was calculated from its modulation transfer function as measured separately under the same detection settings with the knife-edge method.

The experimental and calculated data were quantitatively compared using the mean squared error (MSE) as metric. The MSE is the average of the squared intensity differences. Please note, that the MSEs for different experimental datasets are in general not comparable with each other due to the different overall electron dose. The MSE was minimized under variation of specimen thickness t , depth of dislocation d , total intensity N , beam tilt τ and the exact positions of the dislocation and diffraction pattern center in the experimental data. Specimen thickness t and dislocation depth d were tested for all relevant values with 2 nm steps in thickness and 5 nm steps for depth. The other parameters were numerically minimized for a given set of (t, d) using the Broyden-Fletcher-Goldfarb-Shanno method implemented numerical ~~Python package *scipy* [33, 34]~~Python package *scipy* [34].

All calculations and data processing were performed using ~~Python [35] and the *PyCTEM* toolkit[36]~~Python and the *PyCTEM* toolkit. Further information about calculation times can be found in Supplementary ~~Data-Note~~ Data-Note 4.

DATA AVAILABILITY

~~The data used within this study are available from the corresponding author upon reasonable request~~experimental data generated in this study have been deposited in the Zenodo repository with the identifier “doi:10.5281/zenodo.10458023” (<https://doi.org/10.5281/zenodo.10458023>).

CODE AVAILABILITY

~~The code used within this study is available from the corresponding author upon reasonable request~~A GitHub repository containing the code used in the analysis is available ([https://github.com/niermann/match_qx](https://github.com/niermann/match_qx)).

REFERENCES

- [1] Hull, D. & Bacon, D. *Introduction to Dislocations (5th Ed.)* (Butterworth-Heinemann, 2011).
- [2] Nakamura, S. The roles of structural imperfections in InGaN-based blue light-emitting diodes and laser diodes. *Science* **281**, 956–961 (1998). <https://doi.org/10.1126/science.281.5379.956>.
- [3] Sutton, A. P. & Balluffi, R. W. *Interfaces in crystalline materials* (Clarendon Press, Oxford, 1996).
- [4] Hirsch, P., Cockayne, D., Spence, J. & Whelan, M. 50 years of TEM of dislocations: Past, present and future. *Philosophical Magazine* **86**, 4519–4528 (2006). <https://doi.org/10.1080/14786430600768634>.
- [5] Ludwig, W. *et al.* Three-dimensional imaging of crystal defects by ‘topo-tomography’. *Journal of Applied Crystallography* **34**, 602–607 (2001). <https://doi.org/10.1107/S002188980101086X>.
- [6] Barnard, J. S., Sharp, J., Tong, J. R. & Midgley, P. A. High-resolution three-dimensional imaging of dislocations. *Science* **313**, 319–319 (2006). <https://doi.org/10.1126/science.1125783>.
- [7] Chen, C.-C. *et al.* Three-dimensional imaging of dislocations in a nanoparticle at atomic resolution. *Nature* **496**, 74 (2013). <https://doi.org/10.1038/nature12009>.
- [8] Rez, P. & Treacy, M. M. J. Three-dimensional imaging of dislocations. *Nature* **503**, E1 (2013). <https://doi.org/10.1038/nature12660>.
- [9] Weyland, M. & Midgley, P. Electron Tomography in *Transmission Electron Microscopy: Diffraction, Imaging, and Spectrometry* (C. B. Carter and D. B. Williams, Eds.), 343–376 (Springer International Publishing, 2016). https://doi.org/10.1007/978-3-319-26651-0_12.
- [10] Gilgenbach, C., Chen, X., Xu, M. & LeBeau, J. Three-dimensional Analysis of Nanoscale Dislocation Loops with Multislice Electron Ptychography. *Microscopy and Microanalysis* **29**, 286–287 (2023). <https://doi.org/10.1093/micmic/ozad067.132>.
- [11] Yang, H. *et al.* Imaging screw dislocations at atomic resolution by aberration-corrected electron optical sectioning. *Nature Communications* **6**, 7266 (2015). <https://doi.org/10.1038/ncomms12532>.

- [12] De Graef, M. *Introduction to Conventional Transmission Electron Microscopy* (Cambridge University Press, 2003). <https://doi.org/10.1017/CB09780511615092>.
- [13] Williams, D. B. & Carter, C. B. *Transmission Electron Microscopy - A Textbook for Materials Science* (Springer, 2009).
- [14] Tanaka, M., Terauchi, M. & Kaneyama, T. *Convergent-Beam Electron Diffraction II* (Jeol LTD., 1988).
- [15] Ophus, C. Four-dimensional scanning transmission electron microscopy (4D-STEM): From scanning nanodiffraction to ptychography and beyond. *Microscopy and Microanalysis* **25**, 563–582 (2019). <https://doi.org/10.1017/S1431927619000497>.
- [16] Howie, A. & Basinski, Z. S. Approximations of the dynamical theory of diffraction contrast. *The Philosophical Magazine: A Journal of Theoretical Experimental and Applied Physics* **17**, 1039–1063 (1968). <https://doi.org/10.1080/14786436808223182>.
- [17] Lubk, A. *et al.* Dynamic scattering theory for dark-field electron holography of 3D strain fields. *Ultramicroscopy* **136**, 42 (2014). <https://doi.org/10.1016/j.ultramic.2013.07.007>.
- [18] Meißner, L., Niermann, T., Berger, D. & Lehmann, M. Dynamical diffraction effects on the geometric phase of inhomogeneous strain fields. *Ultramicroscopy* **207**, 112844 (2019). <https://doi.org/10.1016/j.ultramic.2019.112844>.
- [19] Koprucki, T., Maltsi, A. & Mielke, A. Symmetries in transmission electron microscopy imaging of crystals with *Proceedings of the Royal Society A: Mathematical, Physical and Engineering Sciences* **478**, 20220317 (2022). <https://doi.org/10.1098/rspa.2022.0317>.
- [20] Hirth, J. & Lothe, J. *Theory of Dislocations (2nd Ed.)* (John Wiley & Sons, New York, 1982).
- [21] Eshelby, J., Read, W. & Shockley, W. Anisotropic elasticity with applications to dislocation theory. *Acta Metallurgica* **1**, 251–259 (1953). [https://doi.org/10.1016/0001-6160\(53\)90099-6](https://doi.org/10.1016/0001-6160(53)90099-6).
- [22] ~~Koprucki, T., Maltsi, A. & Mielke, A. Symmetries in transmission electron microscopy imaging of crystals with *Proceedings of the Royal Society A: Mathematical, Physical and Engineering Sciences* **478**, 20220317 (2022).~~
- [23] Ponce, F. & Bour, D. Nitride-based semiconductors for blue and green light-emitting devices. *Nature* **386**, 351–359 (1997). <https://doi.org/10.1038/386351a0>.
- [24] Chien, F. R. *et al.* ~~Growth defects in GaN films on 6H-SiC substrates~~Growth defects in GaN films on 6H-SiC *Applied Physics Letters* **68**, 2678–2680 (1996). <https://doi.org/10.1063/1.116279>.

- [25] Lehmann, M. & Lichte, H. Tutorial on off-axis electron holography. *Microscopy and Microanalysis* **8**, 447–466 (2002). <https://doi.org/10.1017/S1431927602020147>.
- [26] Mkhoyan, K. A., Maccagnano-Zacher, S. E., Thomas, M. G. & Silcox, J. Critical role of inelastic interactions in quantitative electron microscopy. *Phys. Rev. Lett.* **100**, 025503 (2008). <https://doi.org/10.1103/PhysRevLett.100.025503>.
- [27] Mendis, B. An inelastic multislice simulation method incorporating plasmon energy losses. *Ultramicroscopy* **206**, 112816 (2019). <https://doi.org/10.1016/j.ultramic.2019.112816>.
- [28] Spence, J. C. H. & Zuo, J. M. *Electron Microdiffraction* (Springer New York, NY, 1992).
- [29] Press, W. H., Teukolsky, S. A., Vetterling, W. T. & Flannery, B. P. *Numerical Recipes in C (2nd Ed.): The Art of Scientific Computing* (Cambridge University Press, USA, 1992).
- [30] Weickenmeier, A. & Kohl, H. Computation of absorptive form factors for high-energy electron diffraction. *Acta Cryst. A* **47**, 590 (1991). <https://doi.org/10.1107/S0108767391004804>.
- [31] Maltsi, A., Niermann, T., Streckenbach, T., Tabelow, K. & Koprucki, T. Numerical simulation of TEM images for In(Ga)As/GaAs quantum dots with various shapes. *Optical and Quantum Electronics* **52**, 257 (2020). <https://doi.org/10.1007/s11082-020-02356-y>.
- [32] Niermann, T., Lubk, A. & Röder, F. A new linear transfer theory and characterization method for image detectors. Part I: Theory. *Ultramicroscopy* **115**, 68 (2012). <https://doi.org/10.1016/j.ultramic.2012.01.012>.
- [33] ~~Scipy. URL <https://scipy.org/>~~
- [34] Nocedal, J. & Wright, S. J. *Numerical Optimization* (Springer New York, NY, 2006). <https://doi.org/10.1007/978-0-387-40065-5>.
- [35] ~~Python. URL <https://www.python.org> .~~
- [36] ~~Niermann, T. URL <http://www.pyctem.org>.~~

ACKNOWLEDGEMENTS

~~We thank Sören Selve for providing us the Aluminum sample.~~ This work was funded by the Deutsche Forschungsgemeinschaft (DFG, German Research Foundation) within projects 492463633 ~~and 403371556~~ ~~(L.N.)~~ ~~and 403371556 (M.L.)~~. ~~We thank Sören Selve for providing us the Aluminum sample.~~

~~AUTHORS'~~AUTHOR CONTRIBUTIONS

L.N. and T.N. conceived the idea and designed the experiment. L.N. conducted the experiment and analyzed the data. T.N. conducted the simulations and developed the algorithms used for simulation and data analysis. L.N. and T.N. wrote the paper. M.L. contributed suggestions and revised the manuscript. All authors read and commented on the manuscript.

COMPETING INTERESTS

The authors declare no competing interests.

~~SUPPLEMENTARY INFORMATION~~

~~Supplementary Information is available for this paper.~~

FIGURE LEGENDS

FIGURES

Figure 1. ~~Dataset overview: (a) scheme~~ Dataset overview. a Scheme of the ~~scanning-acquisition~~ and ~~evaluating-evaluation~~ process of the 4D-dataset. The specimen is tilted into a systematic row condition. The electron beam scans in within the (x, y) -plane over the specimen and for each scan position a diffraction pattern is acquired. The direction of the systematic row defines the reciprocal space direction q . The diffraction patterns with the same x -distance to the dislocation are averaged along the perpendicular y -direction. The intensities in these averaged diffraction patterns are further averaged perpendicular to the systematic row-direction (in q' -direction). In this way, for every x -position a q -profile is obtained, which results in intensities $I(q, x)$ within a (q, x) -plane. ~~(b)~~ b Annular dark field (ADF) image in (0002) systematic row conditions with evaluated area marked by red rectangle (the red arrow marks the spatial direction x for dislocation A) ~~(c)~~ Diffraction pattern averaged over all scan coordinates within the red rectangle of a (the red arrow marks the reciprocal space direction q for dislocation A). ~~(d)~~ ADF image in $(2\bar{1}\bar{1}0)$ systematic row conditions of the same region with evaluated area marked by red rectangle (the red arrow marks the spatial direction x for dislocation B), ~~(e)~~ Diffraction pattern averaged over all scan coordinates within the red rectangle of d (the red arrow marks the reciprocal space direction q for dislocation B). The crystal-directions are indicated in images ~~(b,b,d)~~ d. The reflections are indicated in the diffraction patterns ~~(c, e)~~. The diffraction pattern in ~~(e,e)~~ c, e have been flipped and rotated to match the scan coordinate system of ~~(b, d)~~ b, d.

Figure 2. ~~Comparison of (q, x) -planes for dislocation A under (0002) -systematic row condition:~~ Comparison of (q, x) -planes for dislocation A under (0002) -systematic row condition. ~~a)-experimental~~ Experimental intensities, ~~(b)-simulated~~ Simulated intensities for a Burgers vector of $\mathbf{b} = [000\bar{1}]$, ~~(c)-mean~~ Mean squared error map between experimental and calculated intensities for different values of specimen thickness t and dislocation depth d (uncolored points indicate untested points or diverged fits, white cross labels parameters used in b). Profiles through the mean ~~square~~ squared error map can be found in Supplementary ~~Figure 4~~ Fig. 8.

Figure 3. ~~Comparison of (q, x) -planes for dislocation B under $(2\bar{1}\bar{1}0)$ -systematic row condition:~~
~~(Comparison of (q, x) -planes for dislocation B under $(2\bar{1}\bar{1}0)$ -systematic row condition.~~
a) ~~experimental~~ Experimental intensities, (**b**) ~~simulated~~ Simulated intensities for a Burgers vector of $\mathbf{b} = \frac{1}{3}[\bar{1}2\bar{1}0]$, (**c**) ~~mean~~ Mean squared error map between experimental and calculated intensities for different values of specimen thickness t and dislocation depth d (uncolored points indicate untested points or diverged fits, white cross labels parameters used in **b**). Profiles through the mean ~~square~~ squared error map can be found in Supplementary ~~Figure 4~~ Fig. 8.